# Modeling of African population history using *f*-statistics is biased when applying all previously proposed SNP ascertainment schemes

**Pavel Flegontov**[1,2,3☯]*, **Ulaş Işıldak**[2☯¤], **Robert Maier**[1☯], **Eren Yüncü**[2¤], **Piya Changmai**[2], **David Reich**[1,4,5,6]*

**1** Department of Human Evolutionary Biology, Harvard University, Cambridge, Massachusetts, United States of America, **2** Department of Biology and Ecology, Faculty of Science, University of Ostrava, Ostrava, Czechia, **3** Kalmyk Research Center of the Russian Academy of Sciences, Elista, Russia, **4** Department of Genetics, Harvard Medical School, Boston, Massachusetts, United States of America, **5** Howard Hughes Medical Institute, Harvard Medical School, Boston, Massachusetts, United States of America, **6** Broad Institute of Harvard and MIT, Cambridge, Massachusetts, United States of America

☯ These authors contributed equally to this work.
¤ Current address: E.Y., Department of Biological Sciences, Middle East Technical U., Ankara, Turkey; U.I., Leibniz Institute on Aging—Fritz Lipmann Institute (FLI), Jena, Germany.
* pflegontov@gmail.com (PF); reich@genetics.med.harvard.edu (DR)

**Data Availability Statement:** All the genetic data analyzed in the manuscripts were either simulated or published. The software used in this manuscript

## Abstract

*f*-statistics have emerged as a first line of analysis for making inferences about demographic history from genome-wide data. Not only are they guaranteed to allow robust tests of the fits of proposed models of population history to data when analyzing full genome sequencing data—that is, all single nucleotide polymorphisms (SNPs) in the individuals being analyzed—but they are also guaranteed to allow robust tests of models for SNPs ascertained as polymorphic in a population that is an outgroup in a phylogenetic sense to all groups being analyzed. True "outgroup ascertainment" is in practice impossible in humans because our species has arisen from a substructured ancestral population that does not descend from a homogeneous ancestral population going back many hundreds of thousands of years into the past. However, initial studies suggested that non-outgroup-ascertainment schemes might produce robust enough results using *f*-statistics, and that motivated widespread fitting of models to data using non-outgroup-ascertained SNP panels such as the "Affymetrix Human Origins array" which has been genotyped on thousands of modern individuals from hundreds of populations, or the "1240k" in-solution enrichment reagent which has been the source of about 70% of published genome-wide data for ancient humans. In this study, we show that while analyses of population history using such panels work well for studies of relationships among non-African populations and one African outgroup, when co-modeling more than one sub-Saharan African and/or archaic human groups (Neanderthals and Denisovans), fitting of *f*-statistics to such SNP sets is expected to frequently lead to false rejection of true demographic histories, and failure to reject incorrect models. Analyzing panels of SNPs polymorphic in archaic humans, which has been suggested as a solution for the ascertainment problem, has limited statistical power and retains important biases. However,

is publicly available at: https://uqrmaie1.github.io/admixtools/.

**Funding:** P.F., U.I., and P.C. were supported by the Czech Ministry of Education, Youth and Sports (program ERC CZ, project no. LL2103). P.F. and P.C. were supported by the Czech Science Foundation (project no. 21-27624S). P.F. was also supported by a subsidy from the Russian federal budget (project No. 075-15-2019-1879 "From paleogenetics to cultural anthropology: a comprehensive interdisciplinary study of the traditions of the peoples of transboundary regions: migration, intercultural interaction and worldview"). This research was funded by NIH grant HG012287, by the Allen Discovery Center program, a Paul G. Allen Frontiers Group advised program of the Paul G. Allen Family Foundation, by John Templeton Foundation grant 61220, by private gifts from Jean-Francois Clin to D.R. and P.F., and by the Howard Hughes Medical Institute. Computational resources for this work were supplied by the projects "e-Infrastruktura CZ (e-INFRA CZ LM2018140) and "IT4Innovations National Supercomputing Center – LM2015070" supported by the Ministry of Education, Youth and Sports of the Czech Republic. The funders played no role in planning of this work, in interpretation of the results, and in preparation of the manuscript.

**Competing interests:** The authors declare no competing interests.

by carrying out simulations of diverse demographic histories, we show that bias in inferences based on $f$-statistics can be minimized by ascertaining on variants common in a union of diverse African groups; such ascertainment retains high statistical power while allowing co-analysis of archaic and modern groups.

## Author summary

Archaeogenetic research on humans remains heavily biased towards Europe, Central and East Asia due to poor preservation of ancient DNA in hot climate. However, the number of studies focused on the history of African human populations is growing. Due to the DNA preservation problems, using targeted enrichment for selected variable loci is almost unavoidable in archaeogenetic research focused on Africans. Moreover, poor quality of archaeogenetic data makes the analytical toolkit rather limited: it is often restricted to methods based on $f$-statistics, PCA, and *ADMIXTURE*. It is known that $f$-statistics may be biased when they are calculated not on whole-genome data, but on sets of SNPs selected in a non-random way. Although this is common knowledge, biases affecting $f$-statistics on such SNP sets ("ascertainment biases") remain poorly explored in practice, and our study is designed to fill this gap. We investigate biases affecting individual $f_4$-statistics and fits of admixture graph models on simulated and real data, explore dozens of ascertainment schemes, and provide a set of guidelines for minimizing bias. We show that ascertainment bias is particularly strong in situations when several African populations are co-analyzed with non-African and archaic (Neanderthal or Denisovan) human groups.

## Introduction

Archaeogenetics has achieved remarkable progress in the last decade [1,2], with genome-wide data for thousands of ancient humans now being published each year. No region of the world is now inaccessible to archaeogenetic research, although isolation of enough authentic DNA from skeletons excavated in tropical and sub-tropical areas [3] or from Pleistocene individuals [4,5] remains a challenge. For generating usable archaeogenetic data from Africa, targeted enrichment of human DNA on dedicated single nucleotide polymorphism (SNP) capture panels is almost always necessary. A majority of ancient DNA studies on African populations [6–13] relied on a SNP capture panel usually termed "1240K" [14,15], and some studies on Upper Paleolithic humans relied on a supplementary panel ("1000K", comprising transversion polymorphisms found in two Yoruba individuals and transversion polymorphisms in the Altai Neanderthal genome) or on its union with 1240K [4,14], or on standalone 1240K [5]. The 1240K panel was constructed of the following elements: all SNPs on the Human Origins array (itself composed of 13 sub-panels, each ascertained as heterozygous in a single high-coverage human genome [16]), all SNPs on the Illumina 650Y array, all SNPs on the Affymetrix 50k XBA array, and smaller numbers of SNPs chosen for other purposes [14]. The 1240K capture panel is now used routinely for analyzing thousands of ancient humans across the world [1,17], and successor panels including the full set of 1240K sites are now available [18].

$f$-statistics [16,19–22] are one of the most widely used tools for analyzing allele frequency data in population genetics, and especially in archaeogenetics where high-quality data required for many analytical approaches (based on, e.g., site frequency spectra or autosomal haplotypes) are typically unavailable. $f$-statistics are of three types (see Patterson *et al.* [16] and a recent

review by Maier *et al.* [23]): $f_4$, $f_3$, and $f_2$ (the latter two statistics are special cases of the former). The $f_4$-statistic $f_4(A, B; C, D)$ measures correlation in allele frequency differences between populations *A* and *B* and populations *C* and *D* (($p_A − p_B$) × ($p_C − p_D$) for allele frequencies *p*), typically averaged over thousands of biallelic single-nucleotide polymorphisms [16,20,22]. The $f_4$-statistic is identical to the ABBA/BABA statistic, also known as the *D*-statistic [24,25], up to a normalization factor, and is a test for treeness. Statistically significant deviations of this statistic from zero (with standard deviation calculated on jackknife replicates of a SNP dataset split into blocks) suggest that the unrooted tree ((*A*, *B*), (*C*, *D*)) does not fit the data, and the sign of the statistic points to pairs of populations potentially connected by gene flow [26]. The $f_2$-statistic (*A*, *B*) is identical to $f_4(A, B; A, B)$ and can be interpreted as the genetic distance between groups *A* and *B*. The $f_3$-statistic (*outgroup*; *A*, *B*) is identical to $f_4$(*outgroup*, *A*; *outgroup*, *B*) and measures genetic drift shared between groups *A* and *B* so it is often used for finding groups/individuals that are genetically closest to the group of interest. A significantly negative value of the $f_3$-statistic (*target*; *A*, *B*) provides a formal test for the target group being a mixture of ancestry sources related to groups *A* and *B* [16,20,22], although a signal from this "admixture-$f_3$" test can be masked by genetic drift in the target group since admixture. Non-zero $f_4$-statistics can sometimes detect admixture even in a population that has experienced substantial genetic drift.

All *f*-statistics can be expressed as linear sums of other *f*-statistics [16,20,22] and have straightforward interpretations in terms of admixture graph edges [16,20,22] and arrangement of individuals in principal component spaces [22]. $f_4$-statistics form the foundation of *qpAdm* [27,28], a method that is used widely in archaeogenetics [29] for fitting simple admixture models (target group *A* = proxy source *B* + proxy source *C*, etc.) that are not phylogenetically explicit. Admixture graphs are also often fitted to *f*-statistics, namely to all possible $f_2$-, $f_3$-, and $f_4$-statistics for a given set of populations [16,23,26]; see more on admixture graphs below. Considering that *f*-statistics and methods relying on them are very useful and popular in population genetics, and in archaeogenetics in particular, it is important to explore carefully biases in *f*-statistics that may arise due to non-random selection ("ascertainment") of SNP loci for analysis.

Bergström *et al.* [30], relying on high-quality genomic data for present-day humans, showed that $f_4$-statistics including three sub-Saharan African groups and one non-African group, or four sub-Saharan African (hereafter "African") groups can be biased when computed on common SNP panels such as Illumina MEGA, the panel used by Li *et al.* [31], and the Affymetrix Human Origins array [16]. Influence of ascertainment on common population genetic analyses (*ADMIXTURE*, $F_{ST}$) was also demonstrated. However, the bias in $f_4$-statistics including archaic humans and apes was not explored.

Bergström *et al.* [30] found that selecting approximately 1.3M SNPs polymorphic in the group composed of high-coverage archaic human genomes (the Altai and Vindija Neanderthals, the "Denisova 3" Denisovan) effectively eliminated the biases affecting $f_4$-statistics calculated on anatomically modern humans (AMH) and including 3 or 4 sub-Saharan African groups. A similar approach (selecting ca. 814K transversion sites variable between the Altai Neanderthal and Denisovan) was proposed by Skoglund *et al.* [6]. A SNP capture reagent relying on this principle, the *myBaits Expert Human Affinities Kit* "Ancestral 850K" module, became available in 2021 from Daicel Arbor Biosciences (https://arborbiosci.com/genomics/targeted-sequencing/mybaits/mybaits-expert/mybaits-expert-human-affinities/). This module targets approximately 850K biallelic transversion SNPs (autosomal and X-chromosomal) ascertained as polymorphic in the group composed of high-coverage archaic human genomes: the Altai [32], Vindija [33], and Chagyrskaya Neanderthals [34], as well as the "Denisova 3" Denisovan genome [35]. This set of variable sites was shown to yield nearly unbiased $F_{ST}$

values for pairs composed of an African and a non-African group within the Simons Genome Diversity Panel (SGDP) dataset [36], in contrast to the 1240K panel (see a technical note on manufacturer's website: https://arborbiosci.com/wp-content/uploads/2021/03/Skoglund_Ancestral_850K_Panel_Design.pdf).

These recommendations are motivated by a theoretical property of $f$-statistics: if a SNP is the result of a single historical mutation and there has not been natural selection, the statistics are expected to be unbiased if SNPs are either unascertained or ascertained as polymorphic in a population that is an outgroup for all populations being analyzed [16,37], and the results in Bergström *et al.* [30] and in the technical note published on the Daicel Arbor Biosciences product page are consistent with this theoretical property of outgroup ascertainment since in these studies SNPs were ascertained on archaic humans, but only AMH populations were analyzed. The problematic case is non-outgroup ascertainment, that is ascertainment on a population that is co-analyzed with others. A series of papers explored non-outgroup ascertainment affecting measures of population divergence on simulated data and real data for humans and domestic animals [37–47]. However, $D$- and $f$-statistics which have more robustness than other allele frequency-based statistics in many cases [16], were not considered in those studies. Limited exploration of non-outgroup ascertainment schemes was performed on simulated data in publications introducing the $D$- and $f$-statistics, with the conclusion that biases are not noticeable in practice [16,25]. A limitation of those initial studies was that they focused on the robustness of formal tests of admixture such as $f_4$-symmetry tests, and did not consider the effects of ascertainment on statistics expected to have non-zero values, which are heavily used in methods that fit proposed topologies of population relationships to data, whether these are full topologies such as the admixture graphs fitted with the *qpGraph* software [16,23], or partially specified topologies as fitted with the *qpAdm* software [27,28].

Existing recommendations for a bias-free SNP enrichment panel also rely on the assumption that archaic humans are nearly perfect outgroups with respect to all AMH, and the expectation that the low-level archaic admixture in non-Africans [24,48] subsequently carried back into Africa to a small extent [33,49] does not contribute substantial bias. But evidence is accumulating in favor of long-lasting population structure in Africa or introgression from an unsampled deeply-diverging archaic group to a common ancestor of AMH [50–55], and it remains unclear how this complex demographic history affects the performance of archaic ascertainment. Moreover, for outgroup ascertainment to be unbiased from the theoretical perspective, the outgroup (or a closely related population) should not be then co-analyzed with other populations [16,37], and the individuals used for ascertainment should not be used as sole representatives of the respective groups. However, given the paucity of high-coverage archaic genomes [32–35] and the usefulness of archaic or African outgroups for calculation of $f_4$-and $D$-statistics and for constraining search spaces of admixture graph topologies [23], these recommendations are often ignored in published $f$-statistic, *qpAdm*, *qpGraph*, and *TreeMix* analyses (e.g., [4,6,9,12, 56–58]). For instance, archaic individuals are co-analyzed with anatomically modern humans on archaic-ascertained SNPs [4,6] or a Yoruba group is co-analyzed with non-Africans on Yoruba-ascertained SNPs [56,57].

Since outgroup ascertainment that is "clean" from the theoretical point of view is rarely used in practice, and since the statistical power of outgroup ascertainment to reject incorrect models of population history was not investigated, it is reasonable to examine the performance of archaic ascertainment and common SNP panels such as 1240K in situations that are often encountered in practice. A technical development important for the work reported here is the *ADMIXTOOLS 2* package [23], which extends the functionality of the original *ADMIXTOOLS* package [16], enabling bootstrap resampling for most tools and a rapid algorithm for finding optima in complex admixture graph topology spaces. The *ADMIXTOOLS 2* package also

makes calculating millions of $f_4$-statistics and fitting tens of thousands of admixture graphs to data a routine task. These developments, taken together, allow us to explore biased $f$-statistics more systematically and provide more informed guidelines for future studies.

Admixture graphs (fitted to all $f$-statistics for a selected group of populations) are simplified demographic models that represent population history as a bifurcating tree with few pulse-like two-way admixture events "mapped" onto it, and with no parameters specifying effective population sizes or explicit dates. The framework was introduced by Patterson *et al.* [16] for checking if a complex historical model inferred from individual $f$-statistics and other methods fits the totality of $f$-statistic data (see also mathematical definitions in [21]). Another series of methods based on $f$-statistics or very similar data (*TreeMix* [59,60], *MixMapper* [61], *miqo-Graph* [62], *AdmixtureBayes* [63], and *findGraphs* [23]) aims at finding a best-fitting admixture graph (or several graphs) by automated exploration of the topology space, with resulting models often considered as approximations of the true population history (see a review in [23]). As demonstrated by Maier *et al.* [23], the latter approach is deeply problematic: $f$-statistics do not constrain even moderately complex topology spaces (e.g., graphs including 8–9 groups and 4–5 admixture events) well enough, and topologically diverse graphs often fit the data significantly better than true simulated histories. However, the former approach, i.e., admixture graph fitting as an easy sanity check of complex historical scenarios, remains a valid use of the method (e.g., [64–66]). Since absolute quality of model fit and relative fits of alternative topologies are crucial for this method, it is worth exploring if ascertainment bias affects fits of topologically diverse admixture graphs to $f$-statistic data.

## Results

### 1. Empirical analyses: exploration of the effect of ascertainment bias on real data

We assembled a set of diploid autosomal genotype calls for 352 individuals (S1 Table) sequenced at high coverage [36,67], including mostly present-day individuals from the Simons Genome Diversity Project (SGDP), several high-coverage ancient genomes with diploid genotype calls [68,69], and three archaic human genomes: the "Denisova 3" Denisovan [35], Vindija [33] and Altai Neanderthals [32]. Relying on this "SGDP+archaic" dataset, we explored a wide array of ascertainment schemes: 1) A/T and G/C SNPs (henceforth "AT/GC") that are, unlike the other mutation classes, unaffected by biased gene conversion [70], and are also unaffected by deamination ancient DNA damage; 2) random thinning of the unascertained or "AT/GC" sets down to the size approximately equal to that of the 1240K SNP panel if missing data are not allowed on a given population set; 3) the 1240K panel [14]; 4) the 1000K panel composed of 997,780 SNPs comprising all transversion polymorphisms found in two African (Yoruba) individuals sequenced to high coverage and transversion polymorphisms found in the Altai Neanderthal genome [14]; 5) the union of the 1000K and 1240K panels termed 2200K [4]; 6) various components of the 1240K panel (the sites included in the Illumina 650Y and/or Human Origins SNP arrays, sites included exclusively in one of them, and the remaining sites); 7) the largest Human Origins sub-panels–panel 4 ascertained as sites heterozygous in a single San individual, panel 5 ascertained as sites heterozygous in a single Yoruba individual, their union (panels 4+5), and panel 13 including sites where a randomly chosen San allele is derived relative to the Denisovan [16] (abbreviated as, e.g., "HO panel 4"); 8) all sites polymorphic in a group uniting three high-coverage archaic genomes: the"Denisova 3" Denisovan, the Altai and Vindija Neanderthals (this ascertainment scheme is abbreviated in this study as "archaic asc." and is similar to those proposed by Bergström *et al.* [30] and in the technical note published on the Daicel Arbor Biosciences product page); 9) transversion sites variable in

the group comprising these three high-coverage archaic genomes (abbreviated as "archaic asc., transv." or "archaic asc., tv."); 10) restricting to SNPs that have high minor allele frequency (MAF >5%) in the whole "SGDP+archaic" dataset, i.e. high global MAF (abbreviated as "global MAF"); 11) restricting to SNPs having high global MAF combined with taking A/T and G/C SNPs only (abbreviated as "AT/GC, global MAF"); 12) restricting to SNPs that have >5% MAF in a selected African or non-African continental meta-population (abbreviated as, e.g., "AFR MAF"), irrespective of their frequency in the other meta-populations (there are nine such meta-populations in our dataset, and thus nine different ascertainments, see S1 Table); 13) restricting to SNPs that have >5% MAF in a selected continental meta-population, A/T and G/C SNPs only (abbreviated as, e.g., "AT/GC, AFR MAF"). For a list of SGDP-derived SNP sets explored in this study and their sizes in terms of groups, individuals, and SNPs see S2 Table. Although this list is surely not exhaustive, it includes all ascertainment schemes most popular in archaeogenetic publications.

To investigate the influence of SNP ascertainment on the ranking of admixture graph models according to their fits to data, we analyzed real data, considering sets of five populations, and as a way of evaluating the effect of SNP ascertainment on the ability to discriminate among different topologies, tested their fit to all possible admixture graph topologies with two admixture events (32,745 distinct topologies with no fixed outgroup; we considered graphs of this complexity as it was unfeasible to work with exhaustive collections of more complex graphs).

First, we explored such exhaustive collections of admixture graphs for three combinations of groups (Fig 1). Residuals of admixture graph model fits on all sites, on a random subset of them approximately equal to the size of the 1240K set, and on AT/GC sites, are tightly correlated (Pearson's $r$ approaches 1). Residuals of admixture graph models restricted to non-Africans only are also highly correlated on all sites and 1240K sites ($r = 0.95$–$0.99$, Fig 1). In contrast, the worst $f_4$-statistic residuals (WR) for graph models including one archaic human, three African groups, and one African group with ca. 60% of non-African ancestry [67] are poorly correlated on all sites and 1240K sites ($r = 0.31$–$0.35$). WR, also referred to as "admixture graph Z-score", is one of two key metrics of admixture graph fit used in this study: it is the Z-score measuring deviation between observed and expected values of an $f_4$-statistic that is predicted most poorly by the admixture graph being tested [23,26]. WR is measured in standard error (SE) intervals, and, by convention, admixture graphs with WR below 3 SE are considered to fit the data well (see, e.g., [68,71–74]). Thus, WR is typically used in the literature to assess absolute fit of admixture graph models to data (e.g., [68,71–75]), and it is used for model ranking in some cases (strictly speaking, WR is just an approximation of absolute model fit, which is hard to calculate since many $f$-statistics for a given population set are correlated [23]). Log-likelihood score (LL score or simply LL) is another metric that is dependent on deviations of all $f$-statistics (for populations included in the model) from their predicted values and on their covariance, and thus more accurately reflects model fit to data [23,71]. However, unlike WR measured in SE units, LL is not easily comparable across admixture graph complexity classes, population sets, and SNP sets (but comparable across topologies of the same complexity on the same set of SNPs and populations), and thus WR is used as the primary admixture graph fit metric in this study.

These results show that admixture graph fit rankings are severely affected by the 1240K ascertainment if certain population combinations are involved. We considered the possibility that this case of poor correlation was driven by admixture graph topologies that were obviously inconsistent with the data–that is, topologies that could be shown to be inconsistent with the data based on gold standard SNP sets without ascertainment bias. However, the lack of strong correlation for some combinations of populations is not just driven by graphs with poor fits to the data. For example, WRs of admixture graphs fitting the data well (WR <2.5 SE) on a

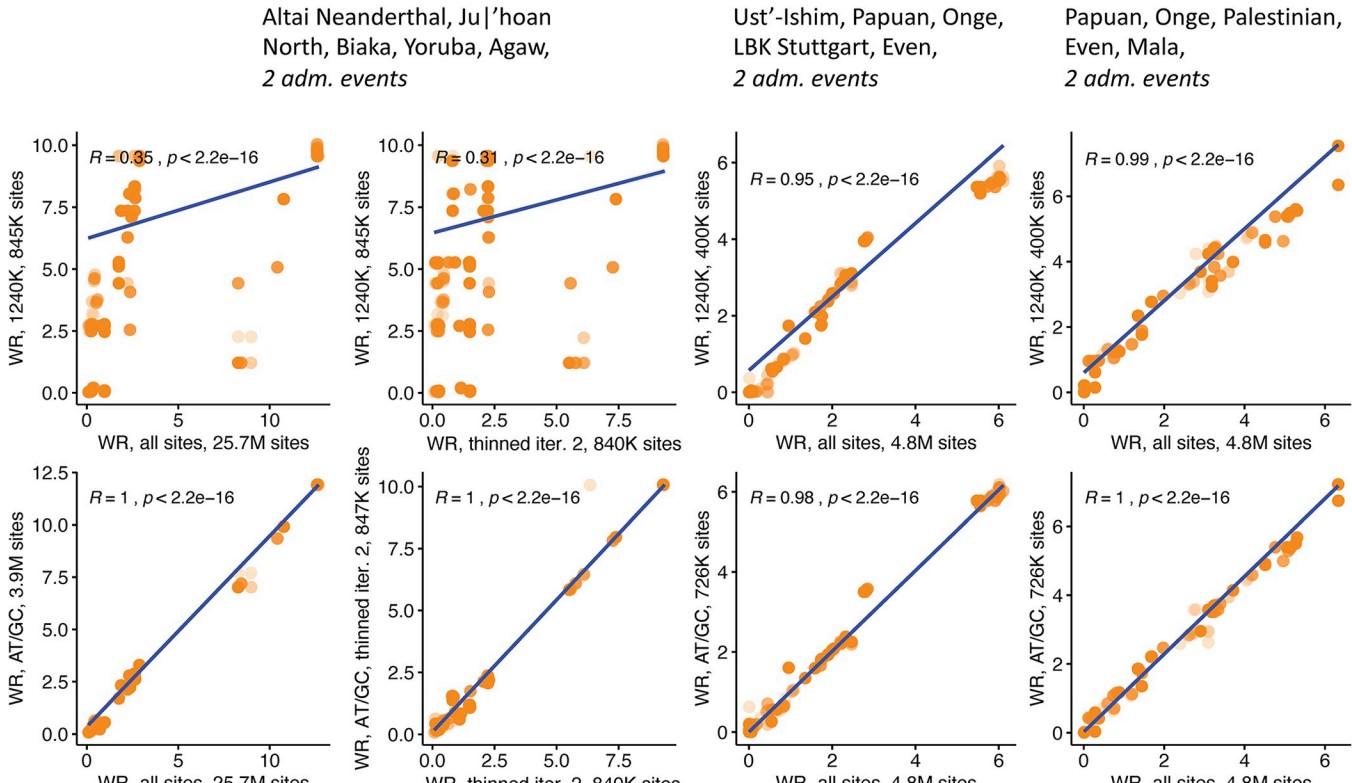

**Fig 1. Scatterplots illustrating the effects of the 1240K ascertainment on worst $f_4$-statistic residuals of admixture graphs (WR), explored on exhaustive collections of simple graphs.** Five thousand best-fitting graphs (according to log-likelihood scores on all sites) of 32,745 possible graphs were selected for each combination of five populations, and correlation of WRs was explored for graphs fitted on all sites and on ascertained datasets. WR, also known as admixture graph Z-score, is the residual of an $f_4$-statistic that is fitted the worst by the admixture graph model. Log-likelihood score of an admixture graph model (LL [23]) reflects deviation of all relevant $f$-statistics from their values predicted under the model and their covariance. Results are shown for three population combinations indicated in plot titles. On x-axes results for all sites or for a randomly thinned site set are shown. Results for the 1240K ascertainment are shown in the upper row on the y-axes, and results for AT/GC sites are shown in the lower row on the y-axes. Linear trends fitted to the plotted points are shown in blue, along with Pearson correlation coefficient.

random subsample of 840,000 sites range from nearly 0 SE to about 10 SE on ca. 845,000 sites included in the 1240K panel (Fig 1). Rejecting a model that fits on unascertained data runs the risk of rejecting the true model, as we show on simulated data in the next section. The converse problem also applies: some admixture graphs are well-fitting (WR <2.5 SE) on the 1240K sites but fit a random sample of sites poorly (WR >5 SE, Fig 1).

Next, we explored the same exhaustive set of admixture graph topologies including five groups and two admixture events on the wider collection of ascertainment schemes listed above and on a larger collection of populations. Twelve combinations of five groups including up to two archaic humans, up to five African groups, and up to five non-African groups were tested. In S1 Fig we compare various ways of looking at the effects of ascertainment on admixture graph fits, using the population quintuplet "Denisovan, Khomani San, Mbuti, Dinka, Mursi" as an example. In Table 1 we focus on the fraction of all graph topologies tested that are considered poorly-fitting under ascertainment (WR >3 SE) but well-fitting on all sites (WR <3 SE) as a metric appropriate for approximately quantifying the most serious effects of ascertainment bias, namely the probability of rejecting the true model. In the supplementary materials, we also show alternative ways of quantifying ascertainment bias: a metric reflecting the statistical power of ascertainment, namely the fraction of all graph topologies tested that are

**Table 1. Performance of ascertainment schemes explored across 12 population quintuplets and assessed as the fraction of all possible admixture graph topologies that are rejected under ascertainment (fit poorly with WR >3 SE) but accepted on all sites (fit well with WR <3 SE).** We also applied the binary classifier to determine if an ascertainment scheme produces unbiased or biased results (the latter cases are highlighted in bold and underlined text). The numbers of population quintuplets or ascertainment schemes affected by bias, or by both bias and low statistical power, are shown in the two rightmost columns and in the two bottom rows, respectively. The level of bias is approximated by the fraction of topologies that are rejected under ascertainment but accepted on all sites, and statistical power is approximated by the fraction of topologies that are, vice versa, accepted under ascertainment but rejected on all sites. The composition of the population sets is shown above the table in an abbreviated way: *arch*, archaic humans, followed by the number of archaic groups in admixture graph models tested; *afr*, Africans, followed by the number of African groups; *nafr*, non-Africans or Africans with substantial non-African admixture [67], followed by the number of such groups. The results for five population quintuplets (listed in the footnote) demonstrating no ascertainment bias are collapsed into one column. The SNP counts correspond to sites polymorphic in larger collections of groups from which the analyzed population quintuplets were taken, see S2 Table. SNP counts vary across the population sets, and minimal and maximal values are shown in separate columns.

| ascertainment type | further details on the ascertainment | min. size of the SNP panel | max. size of the SNP panel | arch 2, afr 3 — Denisovan, Altai, Yoruba, Dinka, Bulala | arch 1, afr 4 — Denisovan, Khomani San, Mbuti, Dinka, Mursi | arch 1, afr 3, nafr 1 — Altai, Ju hoan North, Biaka, Yoruba, Agaw | arch 1, afr 2, nafr 2 — Altai, Ju hoan North, Luhya, Palestinian, Spanish | afr 4, nafr 1 — Bedzan, Cameroon SMA, Esan, Mozabite, Masai | afr 3, nafr 2 — Mbuti, Biaka, Ngumba, LBK, Iranian | afr 1, nafr 4 — Luo, Bedouin B, Jordanian, Abkhasian, Sardinian | afr 5; afr 4, nafr 1; nafr 5 — 5 population quintuplets showing no biased results * | number of biased pop. sets | number of biased pop. sets, both metrics |
|---|---|---|---|---|---|---|---|---|---|---|---|---|---|
| AT/GC | A<>T and G<>C mutations | 805,042 | 1,757,840 | 0.00% | 0.01% | 0.00% | 0.00% | 0.00% | 0.31% | 0.00% | 0.00% | 0 | 0 |
| 1240K | 1240K panel | 501,429 | 663,239 | 0.02% | **1.02%** | **7.97%** | **5.53%** | 0.00% | 0.00% | 0.00% | 0.00% | 3 | 4 |
| 1240K components | Illumina 650Y sites | 256,277 | 304,292 | 0.01% | 0.02% | **7.99%** | **5.53%** | 0.00% | 0.00% | **0.02%** | 0.00% | 2 | 3 |
|  | sites exclusive to Illumina 650Y | 183,680 | 216,478 | 0.00% | 0.01% | **7.97%** | **5.53%** | 0.00% | 0.00% | **0.06%** | 0.00% | 3 | 4 |
|  | sites included in both Illumina 650Y and Human Origins | 72,597 | 87,814 | 0.02% | 0.00% | **7.97%** | **5.53%** | 0.00% | 0.00% | 0.00% | 0.00% | 2 | 6 |
|  | Human Origins sites | 244,922 | 354,460 | 0.00% | 0.00% | **7.95%** | 0.00% | 0.00% | 0.01% | 0.00% | 0.00% | 1 | 3 |
|  | sites exclusive to Human Origins | 171,249 | 266,646 | 0.00% | 0.00% | **6.88%** | 0.00% | 0.00% | 0.00% | 0.00% | 0.00% | 1 | 2 |
|  | 1240K, other sites | 67,096 | 92,301 | 0.00% | **0.00%** | **7.95%** | 0.00% | 0.00% | 0.00% | 0.00% | 0.00% | 1 | 4 |
| Human Origins (HO) panels | panel 13 based on a San individual and Denisovan | 67,557 | 89,655 | 0.00% | 0.17% | **6.99%** | 0.00% | 0.00% | 0.00% | 0.00% | 0.00% | 1 | 6 |
|  | panel 4 based on a San individual | 52,862 | 94,493 | 0.00% | 0.00% | 0.00% | 0.00% | 0.00% | 0.01% | 0.00% | 0.00% | 0 | 4 |
|  | panel 5 based on a Yoruba individual | 44,674 | 73,180 | **0.72%** | 0.00% | **5.25%** | 0.00% | 0.00% | 0.00% | 0.00% | 0.00% | 2 | 8 |
|  | panels 4 and 5 | 46,701 | 157,126 | 0.02% | 0.00% | 0.00% | 0.00% | 0.00% | 0.01% | 0.00% | 0.00% | 0 | 1 |
| 1000K & 2200K | 1000K: transversions in 2 Yoruba ind. and in Altai Neand. | 364,079 | 590,775 | 0.00% | 0.00% | **6.89%** | 0.00% | 0.00% | 0.01% | 0.00% | 0.00% | 1 | 2 |
|  | 2200K panel = 1000K + 1240K panel | 814,915 | 1,190,758 | 0.02% | 0.00% | **7.95%** | 0.00% | 0.00% | 0.01% | 0.00% | 0.00% | 1 | 1 |

*(Continued)*

Table 1. (Continued)

| ascertainment type | further details on the ascertainment | min. size of the SNP panel | max. size of the SNP panel | arch 2, afr 3 (Denisovan, Altai, Yoruba, Dinka, Bulala) | arch 1, afr 4 (Denisovan, Khomani San, Mbuti, Dinka, Mursi) | arch 1, afr 3, nafr 1 (Altai, Ju hoan North, Biaka, Yoruba, Agaw) | arch 1, afr 2, nafr 2 (Altai, Ju hoan North, Luhya, Palestinian, Spanish) | afr 4, nafr 1 (Bedzan, Cameroon SMA, Esan, Mozabite, Masai) | afr 3, nafr 2 (Mbuti, Biaka, Ngumba, LBK, Iranian) | afr 1, nafr 4 (Luo, Bedouin B, Jordanian, Abkhasian, Sardinian) | afr 5, afr 4, nafr 1; nafr 5 (5 population quintuplets showing no biased results *) | number of biased pop. sets | number of biased pop. sets, both metrics |
|---|---|---|---|---|---|---|---|---|---|---|---|---|---|
| archaic asc. | transitions and transversions | 525,014 | 1,555,781 | 10.48% | 0.00% | 6.90% | 0.01% | 0.00% | 0.02% | 0.00% | 0.00% | 2 | 8 |
| | transversions | 165,249 | 484,675 | 10.48% | 0.00% | 6.88% | 0.00% | 0.00% | 0.00% | 0.00% | 0.00% | 2 | 8 |
| MAF | retaining sites with >5% global MAF or: | 2,129,201 | 2,511,335 | 0.00% | 0.00% | 7.95% | 0.00% | 0.00% | 0.00% | 0.00% | 0.00% | 1 | 2 |
| | >5% MAF in Africans unadmixed with non-Africans | 2,045,769 | 3,231,875 | 1.03% | 0.00% | 6.93% | 0.00% | 0.02% | 0.01% | 0.00% | 0.00% | 2 | 2 |
| | >5% MAF in all Africans | 2,109,808 | 3,120,326 | 0.04% | 0.00% | 6.93% | 0.00% | 0.00% | 0.01% | 0.00% | 0.00% | 1 | 1 |
| | >5% MAF in Native Americans | 1,513,207 | 1,764,715 | 0.02% | 0.00% | 7.99% | 5.19% | 0.00% | 0.00% | 0.00% | 0.00% | 2 | 3 |
| | >5% MAF in Central Asians and Siberians | 1,843,262 | 2,150,675 | 0.02% | 0.00% | 7.97% | 0.00% | 0.24% | 0.01% | 0.00% | 0.00% | 1 | 2 |
| | >5% MAF in East Asians | 1,723,831 | 2,020,860 | 0.00% | 0.00% | 7.99% | 5.53% | 0.00% | 0.00% | 0.00% | 0.00% | 2 | 3 |
| | >5% MAF in Europeans | 1,885,336 | 2,192,571 | 0.00% | 0.00% | 7.95% | 0.00% | 0.00% | 0.00% | 0.00% | 0.00% | 1 | 2 |
| | >5% MAF in Middle Eastern groups | 2,018,884 | 2,306,319 | 0.01% | 0.00% | 7.95% | 0.00% | 0.00% | 0.00% | 0.00% | 0.00% | 1 | 1 |
| | >5% MAF in Papuans and Aboriginal Australians | 1,515,022 | 1,791,390 | 1.03% | 0.00% | 8.00% | 5.53% | 0.24% | 0.31% | 0.00% | 0.00% | 3 | 4 |
| | >5% MAF in South Asians | 1,908,459 | 2,235,024 | 0.00% | 0.00% | 7.97% | 0.00% | 0.00% | 0.00% | 0.00% | 0.00% | 1 | 2 |

(Continued)

**Table 1.** (Continued)

| ascertainment type | further details on the ascertainment | min. size of the SNP panel | max. size of the SNP panel | arch 2, afr 3 (Denisovan, Altai, Yoruba, Dinka, Bulala) | arch 1, afr 4 (Denisovan, Khomani San, Mbuti, Dinka, Mursi) | arch 1, afr 3, nafr 1 (Altai, Ju hoan North, Biaka, Yoruba, Agaw) | arch 1, afr 2, nafr 2 (Altai, Ju hoan North, Luhya, Palestinian, Spanish) | afr 4, nafr 1 (Bedzan, Cameroon SMA, Esan, Mozabite, Masai) | afr 3, nafr 2 (Mbuti, Biaka, Ngumba, LBK, Iranian) | afr 1, nafr 4 (Luo, Bedouin B, Jordanian, Abkhasian, Sardinian) | afr 5, afr 4, nafr 1; nafr 5 (5 population quintuplets showing no biased results *) | number of biased pop. sets | number of biased pop. sets, both metrics |
|---|---|---|---|---|---|---|---|---|---|---|---|---|---|
| AT/GC MAF | retaining AT/GC sites with >5% global MAF or: | 323,296 | 378,287 | 0.00% | 0.00% | 7.95% | 0.00% | 0.00% | 0.00% | 0.00% | 0.00% | 1 | 2 |
| | AT/GC, >5% MAF in Africans unadmixed with non-Africans | 309,172 | 486,906 | 0.00% | 0.00% | 6.93% | 0.00% | 0.00% | 0.00% | 0.00% | 0.00% | 1 | 1 |
| | AT/GC, >5% MAF in all Africans | 319,053 | 470,070 | 0.00% | 0.01% | 6.93% | 0.00% | 0.00% | 0.00% | 0.00% | 0.00% | 1 | 1 |
| | AT/GC, >5% MAF in Native Americans | 229,939 | 266,113 | 0.00% | 0.00% | 7.97% | 6.11% | 0.05% | 0.32% | 0.00% | 0.00% | 2 | 3 |
| | AT/GC, >5% MAF in Central Asians and Siberians | 280,103 | 324,245 | 0.00% | 0.00% | 7.97% | 0.00% | 0.25% | 0.05% | 0.00% | 0.00% | 2 | 3 |
| | AT/GC, >5% MAF in East Asians | 261,857 | 304,567 | 0.00% | 0.00% | 7.99% | 0.00% | 0.00% | 0.32% | 0.00% | 0.00% | 1 | 2 |
| | AT/GC, >5% MAF in Europeans | 285,723 | 330,244 | 0.00% | 0.00% | 7.95% | 0.00% | 0.00% | 0.00% | 0.00% | 0.00% | 1 | 3 |
| | AT/GC, >5% MAF in Middle Eastern groups | 306,450 | 347,536 | 0.00% | 0.00% | 7.95% | 0.00% | 0.00% | 0.00% | 0.00% | 0.00% | 1 | 3 |
| | AT/GC, >5% MAF in Papuans and Aboriginal Australians | 230,124 | 272,093 | 1.03% | 0.00% | 8.00% | 5.53% | 0.24% | 0.32% | 0.00% | 0.00% | 3 | 4 |
| | AT/GC, >5% MAF in South Asians | 289,739 | 336,996 | 0.00% | 0.00% | 6.95% | 0.00% | 0.00% | 0.00% | 0.00% | 0.00% | 1 | 3 |
| | number of biased asc. = > | | | 6 | 1 | 33 | 9 | 1 | 0 | 1 | 0 | | |
| | number of biased asc., both metrics = > | | | 6 | 2 | 61 | 9 | 10 | 0 | 5 | 4** | | |

* 1) Mbuti, Baka, Laka, Fulani, Bantu Tswana
2) Khomani San, Bakola, Igbo, Mursi, Aari
3) Australian, Quechua, Mayan, Lezgin, French
4) Papuan, Chipewyan, Eskimo Naukan, Finnish, Sardinian
5) Karitiana, Cree, Eskimo Sireniki, Hungarian, Icelandic

** average number of biased ascertainments per population quintuplet

well-fitting under a given ascertainment (WR <3 SE) but poorly-fitting on all sites (WR >3 SE) (S3 Table), and squared Pearson correlation coefficient for fits (WR or LL) of admixture graphs on unascertained vs. ascertained data (S2 and S3 Figs, S4 and S5 Tables).

Although we recognize that there can be no strict rule for classifying ascertainments into biased and unbiased ones since they form a continuum, for a high-throughput analysis a classifier is useful. Moreover, fits of a given admixture graph model vary even in the absence of ascertainment bias, due to random site sampling effects (Figs 2 and S1), as was shown in previous work [23]. In this study, we considered a SNP set biased if a bias metric (such as the fraction of topologies fitting poorly under ascertainment but well on all sites) was above (or below, as appropriate) the 2.5$^{th}$ percentile of this metric's distribution across 200 sets of randomly sampled SNPs equal to the size of the 1240K set for a given population combination (with no missing data allowed at the group level).

Inspecting the key metric of ascertainment performance (the fraction of topologies that fit poorly under ascertainment but well on all sites), we found only three site sampling schemes that, following the above-mentioned rule, were classified as unbiased for all the population quintuplets tested: AT/GC, HO panel 4, and the union of HO panels 4 and 5 (Table 1). However, due to the small number of sites in the latter two panels, the union of HO panels 4 and 5, and especially panel 4, lack power to reject admixture graph models as compared to the 1240K panel and to AT/GC, as we show in S3 Table. Thus, the only ascertainment scheme that is problem-free according to both metrics is a random one: taking the A/T and G/C mutation classes.

Among the population quintuplets tested, "Altai Neanderthal, Ju|'hoan North, Biaka, Yoruba, Agaw" (Figs 2 and S2A) and "Altai Neanderthal, Ju|'hoan North, Luhya, Palestinian, Spanish" (S2B Fig) are most susceptible to ascertainment bias (Table 1). A very similar quintuplet "Altai Neanderthal, ancient South African hunter-gatherers, Biaka, Yoruba, Agaw" is encountered within more complex admixture graph models that occupy a central place in Lipson's *et al.* [9,12] analyses based on 1240K data (see an investigation of bias affecting the admixture graphs from these studies in S1 Text and S4 and S5 Figs). As explored below on real and simulated data, a class of $f_4$-statistics that are strongly affected by non-outgroup SNP ascertainment underlies admixture graphs for both problematic population quintuplets: $f_4$(African$_x$, archaic; African$_y$, non-African). On the other hand, population sets including no archaic human were virtually unbiased (Table 1), but some ascertainment schemes showed limited power to reject admixture graph models in these cases (S3 Table).

Archaic ascertainment has been suggested in the literature [6,30] as a way to reduce ascertainment bias. However, this approach is guaranteed to work only if the outgroup or a related group is not included itself in admixture graphs or $f$-statistics, and if individuals used for ascertainment are not sole representatives of the respective groups in an analysis. Indeed, we show that archaic ascertainment is biased in the case of the most problematic population quintuplet "Altai Neanderthal, Ju|'hoan North, Biaka, Yoruba, Agaw" (Table 1); in fact, archaic ascertainment is by far the most biased ascertainment approach for population sets including both Neanderthal and Denisovan individuals (Table 1, see also results on simulated data below), and in our analysis it also emerged as the scheme with the lowest statistical power to reject admixture graph models (S3 Table).

If we combine both key metrics of ascertainment performance (the fraction of topologies fitting poorly under ascertainment but fitting well on all sites, and the fraction of topologies fitting well under ascertainment but poorly on all sites), the 1240K and archaic ascertainments are out-performed by many ascertainment schemes, and most notably by the following: 1) HO panels 4+5; 2) the 2200K panel, which combines various kinds of ascertainment such as the 1240K panel, ascertainment on two Yoruba individuals, and on the Altai Neanderthal [14];

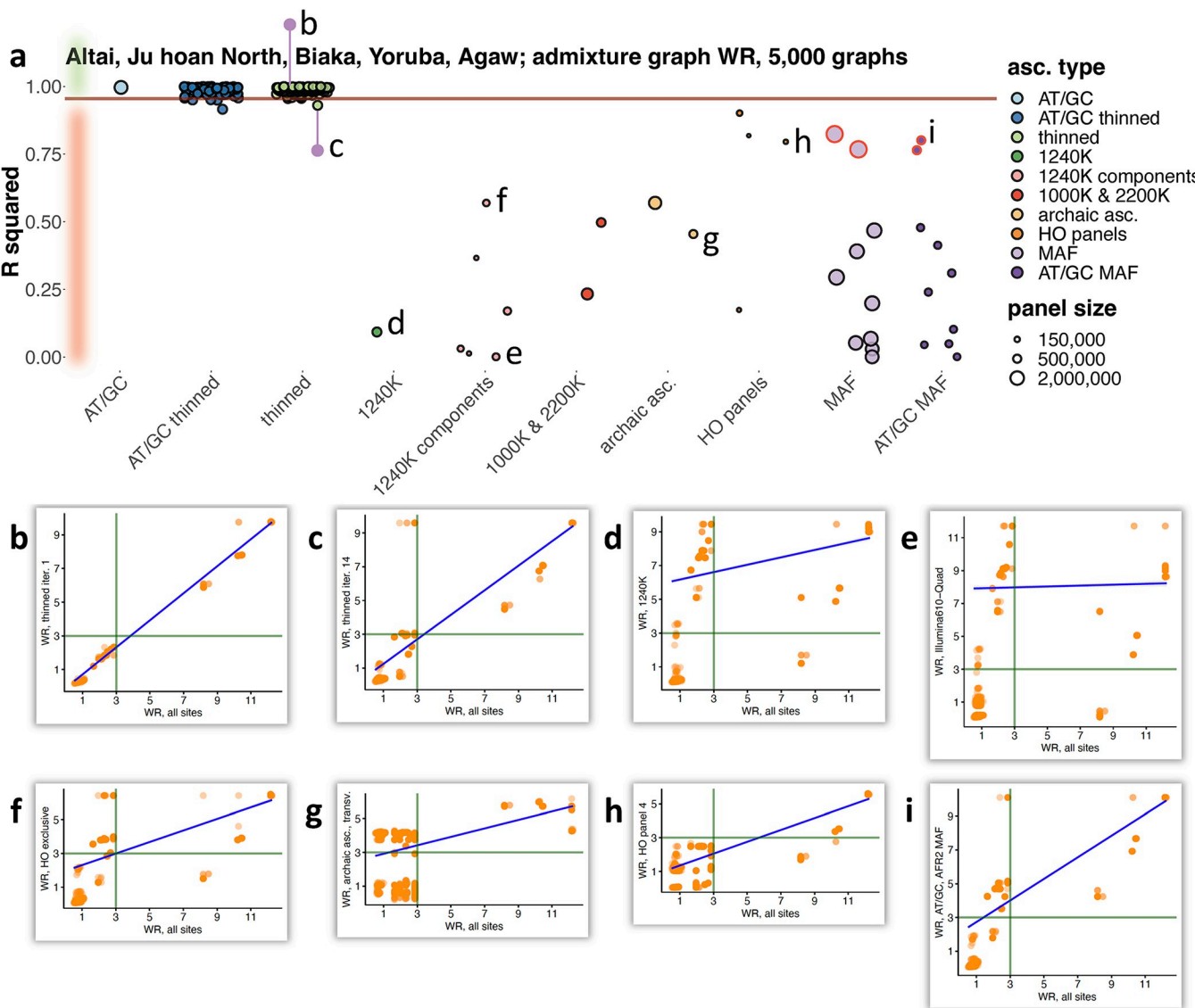

**Fig 2. The effect of ascertainment bias on admixture graph fits illustrated on a population combination "Altai Neanderthal, Ju|'hoan North, Biaka, Yoruba, Agaw".** Five thousand best-fitting graphs (according to log-likelihood scores on all sites) of 32,745 possible graphs were selected, and correlation of worst $f_4$-statistic residuals (WR) of admixture graphs was explored for models fitted on all sites and on ascertained sites. Results for ascertainment on variants common in Africans (either those having no detectable West Eurasian ancestry or all Africans in the SGDP dataset) are circled in red. Thirty eight site subsampling schemes were analyzed (see panel **a**): 1) AT/GC; 2) random thinning of the AT/GC dataset to the 1240K SNP count for a given combination of groups (no missing data allowed at the group level), results for 100 thinned replicates are shown; 3) random thinning of all sites to the 1240K SNP count, results for 100 thinned replicates are shown; 4) the 1240K enrichment panel; 5) major components of the 1240K panel: sites included in the Illumina 650Y and/or Human Origins (HO) SNP arrays, sites included exclusively in one of them, and remaining sites; 6) the 1000K and 2200K SNP panels; 7) archaic ascertainment (either all such sites or transversions only); 8) the largest HO panels (4, 5, 13) or their union (4+5); 9) MAF ascertainment in one of nine continental-scale groups; 10) the same procedure repeated on AT/GC sites. The size of the resulting SNP panels is coded by point size, and the ten broad ascertainment types are coded by color according to the legend. Squared Pearson correlation coefficients ($R^2$) for admixture graph WR on unascertained vs. ascertained data are plotted. The 2.5th WR percentile of all the thinned replicates combined, including those on all sites and AT/GC sites, is marked with the brown line. The area of the plot where ascertainments are considered biased according to this classifier is highlighted in red on the left. Scatterplots illustrating effects of selected ascertainment schemes on WR are shown in panels **b–i**. Dots on these scatterplots correspond to distinct admixture graph topologies. The corresponding ascertainment schemes are marked with letters b–i in panel **a**.

and 3) restricting to variants that are common in the African meta-population in SGDP (abbreviated as "AFR MAF", S1 Table), optionally followed by removal of all mutation classes except for A/T and G/C (Table 1). Squared Pearson correlation coefficient ($R^2$) for admixture graph WR on unascertained vs. ascertained data is in some cases informative in a way that the fractions of poorly/well-fitting topologies are not. As illustrated in Fig 2, $R^2$ may differ substantially across ascertainment schemes while the fractions of topologies fitting poorly under ascertainment but well on all sites or *vice versa* stay nearly constant across most ascertainment schemes (Tables 1 and S3). Considering $R^2$ for admixture graph WR, the AFR MAF scheme emerges as the least biased form of ascertainment (S4 Table). We note that conclusions of this sort are not quantitative since our collection of 12 population quintuplets, although diverse, is just a small sample from the vast set of all possible population combinations. However, exploring all possible combinations is infeasible, and we consider our approach to be useful as a practical guide for assessing the performance of SNP ascertainment when admixture graphs including archaic humans, Africans, and non-Africans are fitted to genetic data.

## 2. Simulation studies confirm the qualitative patterns from exploration of empirical data

A major limitation of our empirical analyses of ascertainment bias is that fitting a model with two admixture events is almost certainly inadequate for the histories relating various sets of five populations being analyzed. Thus, it is almost certain that all fitted models will be wrong. When we fit wrong models, we have no guarantee that the (incorrect) admixture graph fit to the data will give the same signal of deviation for different SNP ascertainments. Different SNP ascertainments including random ascertainments will simply be sensitive to different aspects of the deviations between the wrong model and the true history. Thus, while the poor correlation between model fits on all sites and under different SNP ascertainment schemes for combinations of archaic humans, sub-Saharan Africans, and non-Africans is a potential signal of bias in analyses, it is valuable to analyze data where the truth is known, as is the case for simulations, to provide clear evidence that typical ascertainment schemes can cause false-positive inferences about history.

Using *msprime v.1.1.1* [76], we simulated genetic data (a diploid genome composed of three 100 Mb chromosomes with recombination) that reproduce the $F_{ST}$ values (S6A Fig) observed when comparing AMH groups, AMH and archaic humans, and AMH and chimpanzee [77]. First, ten independent simulations were performed under one admixture graph topology (Fig 3) serving as a case study, and then the analysis was expanded to dozens of random topologies (Fig 4). The former admixture graph (Fig 3A) reproduces some known features of the genetic history of anatomically modern and archaic humans, but differs in other respects from the widely accepted model [32,53]. The Neanderthal gene flow to the ancestors of non-Africans (via an unsampled proxy group) was either simulated or omitted.

We tested several non-outgroup ascertainment schemes (Fig 3): 1) "HO, 1 panel", ascertainment on heterozygous sites in a randomly selected individual from the "African 2" group (this ascertainment follows the scheme used for generating some of the 12 panels of sites comprising the Human Origins SNP array [16]); 2) "HO, 4 panels", ascertainment on heterozygous sites in four randomly selected individuals, one per each "AMH" group (we consider the resulting SNP set to be qualitatively similar to the whole Human Origins SNP set); 3) archaic ascertainment (sites polymorphic in a group composed of one "Denisovan" individual and one individual per each "Neanderthal" group; the same individuals were subsequently used for calculating *f*-statistics); 4) "AFR MAF", that is restricting to sites with MAF >5% in the union of two "African" groups; 5) similar MAF-based ascertainment on two "non-African" groups ("non-AFR MAF") or 6) on all four "AMH" groups ("global MAF").

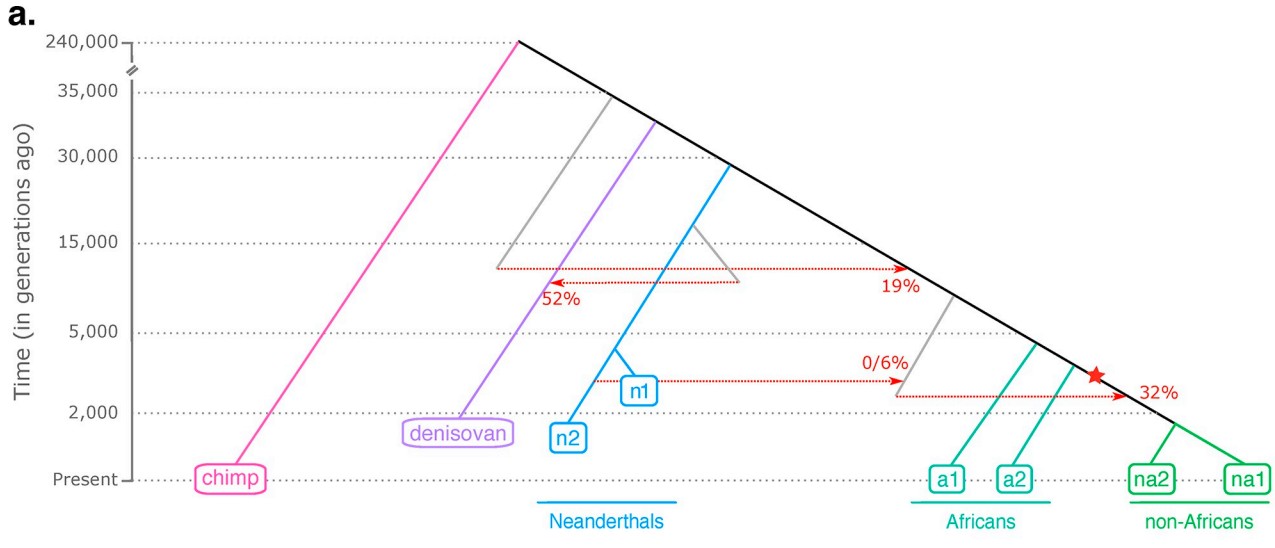

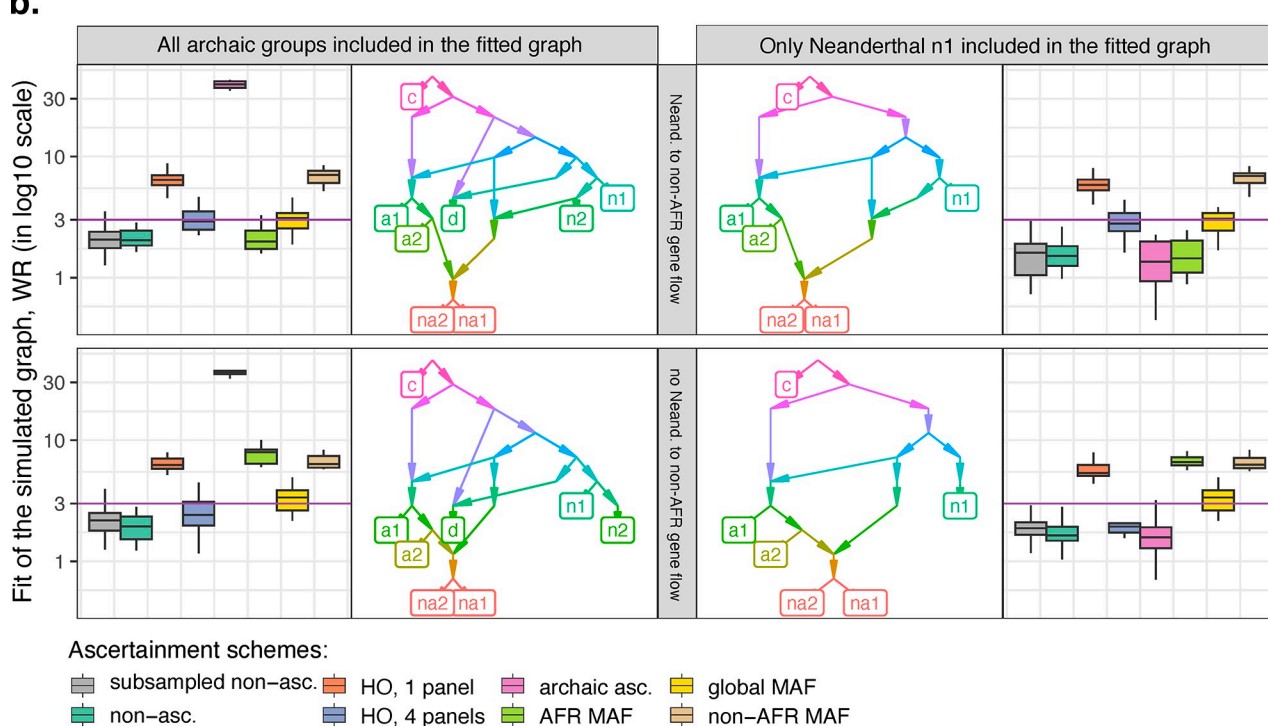

**Fig 3. Exploring the influence of non-outgroup ascertainment on fits of admixture graphs in the case of a single simulated history reproducing some known features of the genetic history of anatomically modern and archaic humans (but differing in other respects from the widely accepted model [53]).** Results are presented for two topologies (with or without the Neanderthal to non-African gene flow simulated) and for eight types of SNP sets: 1) 10 sets of randomly selected variable sites matching the average size of the "HO one-panel" set, 500K sites (abbreviated as "subsampled non-asc."); 2) unascertained sites (on average 5.55M polymorphic sites without missing data at the group level); 3) HO one-panel ascertainment based on the "African 2" group (500K sites on average across simulation iterations); 4) HO four-panel ascertainment, based on randomly selected individuals from four groups ("African 1", "African 2", "non-African 1", and "non-African 2", 1.34M sites on average); 5) archaic ascertainment (1.05M sites on average); 6) "AFR MAF", that is restricting to sites with MAF >5% in the union of the "African 1" and "African 2" groups (1.85M sites on average); 7) global MAF ascertainment on the union of the "African 1", "African 2", "non-African 1", and "non-African 2" groups (1.62M sites on average); 8) non-African MAF ascertainment on the union of the "non-African 1" and "non-African 2" groups (1.48M sites on average). (**a**) The simulated topology, with dates (in generations) shown on the y-axis (for the sake of visual clarity, the axis is not to scale). The Neanderthal to non-African gene flow was simulated either at 0% or at ~2% as shown in the figure. Effective population sizes and population split times are omitted for clarity (see S13 Table). The out-of-Africa bottleneck is marked with a star. (**b**) Boxplots illustrating the effects of various ascertainment schemes on fits (worst $f_4$-statistic residuals, WR) of the correct admixture graphs. The dashed line on the logarithmic scale marks a WR threshold often used in the literature for classifying models into fitting and non-fitting ones, 3 standard errors. The observation that common

ascertainment schemes consistently produce much higher Z-scores than this threshold provides unambiguous evidence that ascertainment bias can profoundly compromise admixture graph fitting. The topologies fitted to the data are shown beside the boxplots. In the panels on the right, simple graphs including only one archaic lineage are fitted (with "Neanderthal 1" used as an example, but very similar results were obtained for the "Neanderthal 2" and "Denisovan" groups). In the panels on the left, results for the full simulated model fitted to the data are shown.

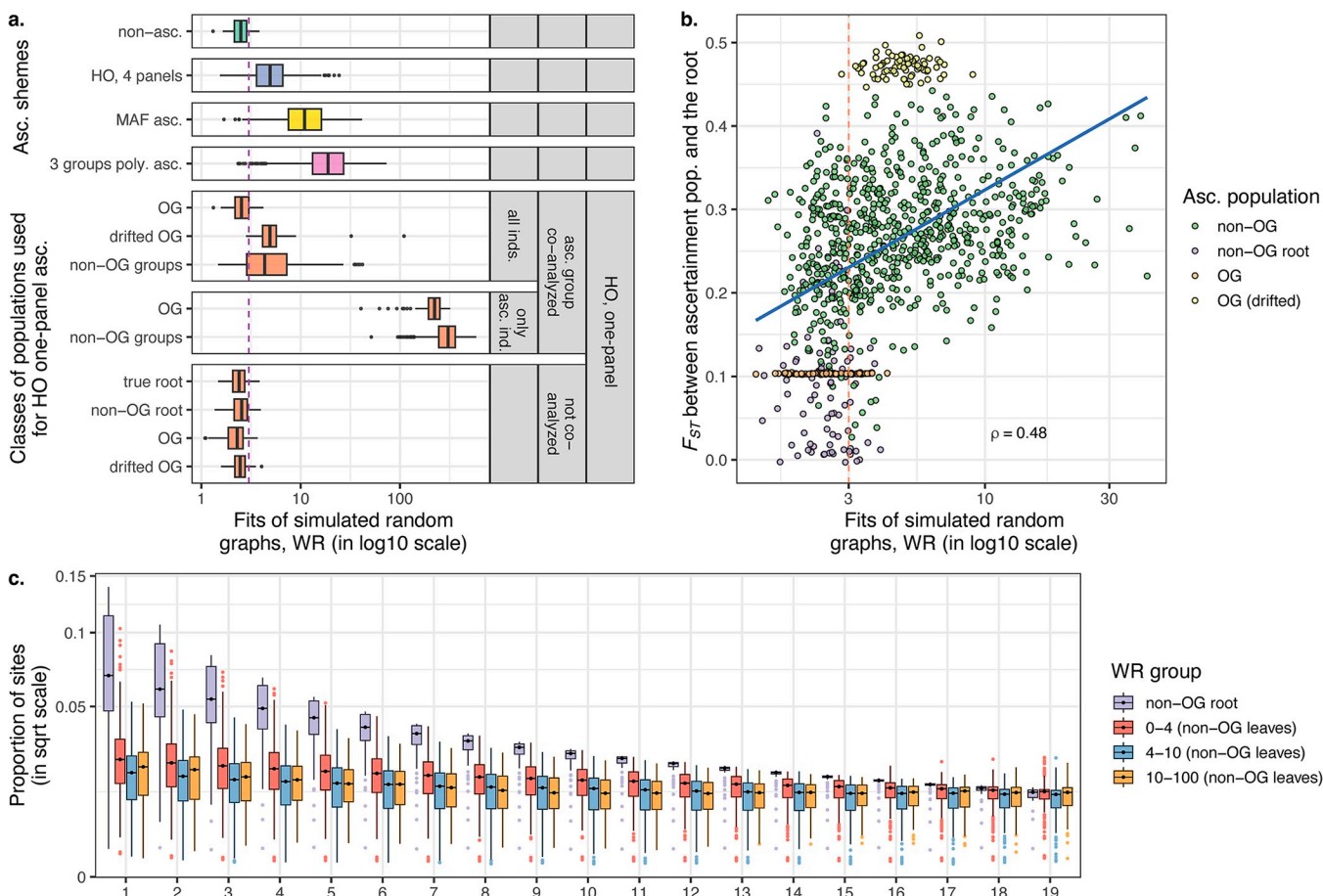

**Fig 4. Effects of non-outgroup and true outgroup ascertainment on fits of admixture graphs explored on a collection of 80 random simulated histories.** The worst $f_4$-statistic residual (WR) of the correct admixture graph was used as a measure of bias. (**a**) WRs for unascertained data and four ascertainment schemes are summarized with boxplots: 1) HO one-panel; 2) HO four-panel; 3) MAF ascertainment on random sets of four populations (abbreviated as "MAF asc."); 4) ascertainment on sites polymorphic in random sets of three individuals (one individual sampled per population; abbreviated as "3 groups poly asc."). HO one-panel ascertainment was performed on various types of simulated populations: on non-outgroup populations ("non-OG groups") or on more or less drifted phylogenetic outgroups (having effective population sizes of 1,000 or 100,000 diploid individuals, respectively) co-modelled with the other populations (abbreviated as "drifted OG" and "OG", respectively). The individual that was used for HO one-panel ascertainment either acted as the only representative of its group for model fitting, or the whole group of 10 individuals was included in the fitted graph. Alternatively, the group used for HO one-panel ascertainment was not included in the fitted graph: it was either the more or less drifted outgroup, the true root, or the last common ancestor of non-outgroup populations ("non-OG root"). These details are reflected in the plot labels on the right and on the y-axis. The dashed vertical line corresponds to WR = 3 SE. (**b**) Correct admixture graphs under HO one-panel ascertainment are guaranteed to be well-fitting (WR < ca. 4 SE) if $F_{ST}$ between the whole population sample used for ascertainment vs. the sample at the root of the simulation is below 0.12. (**c**) DAF spectra (derived allele count in a sample of 20 chromosomes vs. proportion of sites in square root scale) are summarized across simulated non-outgroup populations binned by the level of bias in admixture graph fits (approximated by WR of the true admixture graph) observed when HO ascertainment is performed on the respective population. DAF spectra in populations that are the last common ancestors of non-outgroup populations (abbreviated as "non-OG root") are shown for comparison. The spectra shown here are based only on sites polymorphic in a sample of 20 chromosomes drawn at the root of the simulation (data for sites that are fixed derived or fixed ancestral are not shown, see complete results in S8 Fig). The boxplots summarize DAF across all the simulated admixture graph topologies, and are binned by derived allele counts from 1 to 19.

First, we fitted the correct admixture graph as often practiced in the literature (e.g., [9,12,78]): including an "ape" outgroup, only one "archaic" individual, and all "AMH" groups. HO one-panel ascertainment always leads to rejection of the correct model in this case, both in the absence and in the presence of the Neanderthal gene flow to non-Africans, with WR ranging from 3.4 to 8.8 SE (Fig 3B). HO four-panel ascertainment is less problematic but led to rejection of the correct model (WR >3 SE) in 9 of 30 cases (in the presence of the Neanderthal gene flow to non-Africans), with WR up to 4.6 SE. Only the archaic and AFR MAF non-outgroup ascertainments (in the presence of the Neanderthal gene flow to non-Africans) did not lead to rejection of these simplified graph topologies, known to be correct since we simulated them. However, when the full simulated model (with the Neanderthal gene flow to non-Africans) including the outgroup and three "archaic" lineages is fitted to the data, all non-outgroup ascertainment schemes become problematic, except for the AFR MAF ascertainment (Fig 3B).

Next, we moved beyond fitting only one true admixture graph to ascertained data and used the exhaustive approach for exploring the stability of model ranking under ascertainment that was applied to the real data above. All possible graph topologies with two admixture events (32,745) were fitted to population quintuplets of the following composition: "Denisovan or Neanderthal 1 or Neanderthal 2", "African 1", "African 2", "non-African 1", "non-African 2". The fractions of all topologies tested that were rejected/accepted under ascertainment but accepted/rejected on all sites (and the bias classifier) were then used to reveal simulation iterations and ascertainment schemes that demonstrated biased model fits (S6 Table). When no Neanderthal/non-African gene flow was simulated, only the non-AFR MAF ascertainment emerged as problematic (at least half of simulation iterations for at least one population quintuplet were classified as affected by bias) according to the fraction of topologies rejected under ascertainment but accepted on all sites (S6 Table). When the Neanderthal to non-African gene flow was simulated, all ascertainment schemes, except for the HO four-panel ascertainment, emerged as problematic according to the same metric (S6 Table). Summarizing these results on model ranking and on fits of the true model, we note that the HO four-panel ascertainment is relatively problem-free in this case study on one simulated topology, unlike archaic, MAF, and HO one-panel ascertainment, but it still led to rejection of the true model more often than on all sites or on random site subsamples (Fig 3B).

Finally, we explored non-outgroup ascertainment schemes that are similar to those presented in Fig 3B but are based on randomly chosen groups (see Methods for details) and were applied to SNP sets resulting from simulated genetic histories in the form of random admixture graphs. Graphs of four complexity classes including 9 or 10 populations and 4 or 5 admixture events were simulated using *msprime v.1.1.1*. Only simulations where pairwise $F_{ST}$ for groups were in the range characteristic for anatomically modern and archaic humans were selected for further analysis, resulting in 20 random topologies per graph complexity class, each including an outgroup (see examples of the simulated histories and $F_{ST}$ distributions in S7 Fig). Fits of the true admixture graph (WR) including an outgroup were compared on all sites and on ascertained SNP sets for each topology and ascertainment iteration (Fig 4A). We note that our simulation setup generated groups sampled at different dates in the past (from 0 to ca. 40,000 generations), and thus widely different levels of genetic drift with respect to the root (Fig 4B).

As illustrated by distributions of true admixture graph WRs in Fig 4A, 'blindly' ascertaining on individuals or sets of groups randomly sampled across the graph almost guarantees rejecting the true historical model by a wide margin. Ascertainment on sites polymorphic in randomly composed sets of three individuals (one individual per group) and restricting to variants common (MAF >5%) in randomly composed sets of four populations are two forms of ascertainment that are especially problematic (Fig 4A). One-panel and four-panel HO

ascertainments often yield acceptable results (WR <3 SE), although median WR of the true graphs equals 4.6 SE for these ascertainment schemes across all graph topologies and all (non-root and non-outgroup) populations used for ascertainment (Fig 4A).

An illuminating result is that $F_{ST}$ between the population used for Human Origins-like ascertainment and the root of the simulation influences WR of the true graph: all ascertainments with $F_{ST} < 0.12$ produce relatively unbiased fits of true graphs (WR <4 SE, see Fig 4B). In other words, ascertainment on heterozygous sites in a single individual taken from a population that is not an outgroup and is co-analyzed with other populations, but is relatively undrifted genetically compared to the root of the simulation, is unbiased, unlike ascertainment on a single individual from a more drifted population. We directly illustrate this effect by comparing results on outgroups co-analyzed with other populations that are more or less drifted with respect to the root (with effective population sizes differing by two orders of magnitude) (Fig 4A and 4B). We also show that co-analyzing the individual used for ascertainment with other groups does not exacerbate bias if that individual is a part of a wider population of 10 individuals. However, if that individual is the only representative of its group for model fitting, WRs are inflated drastically (Fig 4A). We finally illustrate the difference between ascertainment on an outgroup that is not co-modelled with the other groups (true outgroup ascertainment) and ascertainment on a phylogenetic outgroup that is included in the fitted model (Fig 4A). The former ascertainment is indeed unbiased even for highly drifted outgroups (Fig 4A), while the latter is not [16,37].

Another way of looking at this phenomenon is through derived allele frequency (DAF) spectra. Ascertainment schemes resulting in relatively unbiased fits of true models (WR <4 SE, Fig 4C) are most often based on populations where the DAF spectrum of sites that were polymorphic at the root is preserved relatively well (see a full version of this plot in S8 Fig). We note that some ascertainments may be unbiased with respect to the true graph but may have low power to reject incorrect admixture graph models due to the paucity of sites with high MAF in "present-day" populations. Indeed, ascertainments on the root itself or on groups genetically close to the root (such as outgroups with a large effective size) are unbiased (Fig 4A and 4B), but on average demonstrated lower power to reject incorrect models as compared to HO one-panel ascertainments on more drifted groups (S9 Fig). Summarizing these observations on a range of random simulated histories, we expect that it is difficult to find an ascertainment scheme that is optimal (at least for the purpose of admixture graph fitting), that is, demonstrates both low bias when testing the true model and high power to reject incorrect models. Our observations on the real data in the preceding section agree with this expectation.

Our results on randomized ascertainment schemes (not to be confused with random site sampling) and simulated histories in the form of random admixture graphs show that ascertainment on groups that are highly drifted with respect to the root of the groups being co-analyzed is problematic. Thus, if proper outgroup ascertainment is impractical (if an outgroup shares few polymorphisms with the other populations analyzed, or if an outgroup is needed for constraining the analysis), unascertained or randomly sampled sets of sites should be treated as a gold standard for admixture graph inference. The 1240K ascertainment is much more complex [14,15] than the ascertainment schemes we explored on simulated data, but its effects are possibly intermediate between the effects of a MAF-based ascertainment (since all common SNP panels are more or less depleted for rare variants) and ascertainment on heterozygous sites in single individuals from several groups (since approximately half of the 1240K sites are derived from the Human Origins SNP array ascertained this way [14]). Thus, we expect an accurate admixture graph including at least one archaic human, at least two African groups, and at least one non-African group (Fig 3A) to fit the data poorly under the 1240K ascertainment.

We also checked if non-outgroup ascertainment could bias the simplest cladality tests in the absence of gene flow. $f_4$-statistics are tests for treeness that are essentially the same [16] as the ABBA-BABA test (*D*-statistic [24,25]) which was used to detect Neanderthal admixture in non-Africans [24]. A tree of four groups conforming to the $f_4$-statistic (*A, B; C, O*) was simulated using *msprime v.1.1.1*, with a tree depth of 4,000 generations (S10A Fig). All the groups had a uniform effective population size of 100,000 diploid individuals, except for a 10x to 10,000x size reduction immediately after the *A-B* divergence (1,999 generations in the past). While the dramatic drop in the effective population size of group A yields a complex shape of the derived allele frequency spectrum in {*A, B*} [79], two of three ascertainment schemes explored here (HO one-panel and MAF ascertainment, but not removal of the derived end of the allele frequency spectrum; see Methods) increase the noise in the $f_4$-statistic (*A, B; C, O*), but do not shift the statistic away from its expectation at 0 (S10 Fig). These results confirm an observation by Patterson *et al*. [16] that in the case of perfect trees non-outgroup SNP ascertainment does not lead to false rejection of cladality. However, as demonstrated in Figs 3 and 4, non-outgroup ascertainment is generally problematic in the case of complex demographic histories with multiple admixture events. This is due to biases on *f*-statistics that have non-zero expected values, which are not relevant to the $f_4$-symmetry test but are very important in admixture graph fitting.

## 3. An overview of $f_4$-statistic biases caused by non-outgroup ascertainment

We explored various classes of $f_4$-statistics exhaustively to obtain a "bird's-eye view" of ascertainment biases that was previously difficult to obtain due to technical challenges in calculating millions of *f*-statistics [30]. Another motivation for this analysis was the fact that it is infeasible to explore fits of large collections of admixture graphs on thousands of population sets, ascertainment schemes and random site subsamples. However, if an exhaustively sampled class of *f*-statistics is demonstrated to be unbiased, all admixture graph fits based on those statistics are expected to be unbiased too.

For this analysis we used residual standard deviation ("residual SE") of a linear trend as a way of measuring correlation between $f_4$-statistic Z-scores on all sites and under ascertainment. We found this metric more convenient than the squared Pearson correlation coefficient ($R^2$) since it is expressed in the same units as Z-scores and thus is an intuitive way of representing deviation of $f_4$-statistic sets on ascertained data from those on unascertained data. We note that it reflects both bias introduced by ascertainment and variance generated by random site sampling. In S7 Table and S11 and S12 Figs we show residual SE values for a collection of 27 exhaustively sampled $f_4$-statistic classes and for the large collection of ascertainment schemes introduced in section 1. The $f_4$-statistic classes explored here can be described concisely as African(all SGDP populations)$_x$;archaic$_y$;chimpanzee$_1$, African(unadmixed with West Eurasians)$_x$;archaic$_y$;Mediterranean/Middle Eastern (abbreviated as Med/ME or ME)$_z$, African (unadmixed with West Eurasians)$_x$;East Asian$_y$(y>0), American/Siberian$_x$;European$_y$;Papuan$_z$. Here, *x*, *y*, and *z* stand for the number of groups in the population quadruplet; thus, "African$_3$; East Asian$_1$" would mean three Africans and one East Asian. All possible distinct $f_4$-statistics composed of those "ingredients" were considered. Although this analysis is by no means exhaustive, the chosen classes of $f_4$-statistics underly a very wide array of admixture graph and *qpAdm* models from the archaeogenetic literature.

The effects of SNP ascertainment vary dramatically across the classes of $f_4$-statistics, but ascertainment schemes based on one or two African individuals (HO panels 4, 5, 13, 4 and 5 combined), on the three archaic individuals (either all sites or transversions only), and components of the 1240K panel such as Illumina 650Y emerged as the worst-performing when results

across all the $f_4$-statistic classes were considered (S7 Table, S11 Fig). Ascertainment schemes based on a global MAF threshold or on a MAF threshold in a single non-African continental meta-population, and the 1000K and 2200K panels are similar in their effects to the 1240K ascertainment (S7 Table, S11 Fig). We recognize that there is a continuum between unbiased and biased ascertainment schemes, and that for nearly all schemes and $f_4$-statistic classes a majority of statistics remain unaffected by ascertainment, but for describing our results in a concise way and for partially factoring out effects of SNP panel size, we applied the criterion similar to that employed above for admixture graph fits: residual SE for an $f_4$-statistic class is higher than the 97.5th percentile across 200 randomly thinned datasets matching the 1240K panel in size. According to this criterion, the 1240K ascertainment is problematic in the case of the following nine $f_4$-statistic classes (S7 Table): 1) $f_4$(African$_w$, African$_x$; African$_y$, African$_z$), 2) $f_4$(African$_x$, African$_y$; African$_z$, Med/ME), 3) $f_4$(African$_x$, African$_y$; African$_z$, East Asian), 4) $f_4$(African$_x$, African$_y$; African$_z$, archaic), 5) $f_4$(African$_x$, African$_y$; African$_z$, chimpanzee), 6) $f_4$(African$_x$, archaic; African$_y$, non-African), 7) $f_4$(African$_x$, archaic; African$_y$, chimpanzee) and $f_4$(African$_x$, African$_y$; archaic, chimpanzee), 8) $f_4$(archaic$_x$, archaic$_y$; archaic$_z$, Med/ME), 9) $f_4$(archaic$_x$, archaic$_y$; archaic$_z$, African), and unproblematic for the remaining 18 classes exhaustively explored in this analysis. Unlike all the other classes explored here (S7 Table, S11 Fig), statistics of the form $f_4$(African$_x$, archaic; African$_y$, non-African) are substantially biased under all types of SNP ascertainment (Fig 5A). The classes $f_4$(African$_x$, African$_y$; African$_z$, X) are problematic under all ascertainment schemes except for AFR MAF (S7 Table, see an example in Fig 5B), and the class $f_4$(African$_w$, African$_x$; African$_y$, African$_z$) is problematic under all ascertainment schemes except for the 1000K, 2200K, and AFR MAF (S7 Table). Scatterplots underlying these residual SE estimates are also shown in Fig 6 (for some of the most problematic classes highlighted above) and in S13–S15 Figs (for all classes). Importantly, the classes of statistics most affected by ascertainment ($f_4$(African$_x$, archaic; African$_y$, non-African), $f_4$(African$_x$, archaic; African$_y$, chimpanzee), $f_4$(African$_x$, African$_y$; African$_z$, X), and $f_4$(African$_w$, African$_x$; African$_y$, African$_z$)) are often relevant for fitting admixture graph models of African population history (see S1 Text). However, for most classes of $f_4$-statistic Z-scores that were classified as problematic, i.e., if residual SE of a linear trend under ascertainment exceeds residual SE under random site sub-sampling to the size of the 1240K panel, absolute residual SE values are below 1 SE (S7 Table and S11 Fig), and thus these statistics are probably not problematic in practice.

If we consider $f_4$-statistics instead of their Z-scores and $R^2$ instead of residual SE, results remain virtually the same (S8 Table). For additional details on $f_4$-statistic classes see S2 Text (and S12–S15 Figs), and for a dissection of effects of ascertainment on few selected $f_4$-statistics see S3 Text (and S9–S12 Tables). In contrast to the $f_4$-statistic classes relevant for modelling African population history, $f_4$-statistics including non-Africans only, or one or two African groups and non-Africans (see an example in Figs 5C and 6), are unproblematic under the 1240K, 2200K, AFR MAF and other MAF-based ascertainments (but demonstrate increased variance due to paucity of sites with high MAF under some other ascertainment types such as HO panels 4 & 5 and archaic ascertainment, S7 Table).

The AFR MAF ascertainment (restricting to variants common across 68 African individuals with little or no Holocene-era ancestry derived from West Eurasians, or across 94 African individuals, S1 Table), emerged as the best-performing non-outgroup ascertainment scheme. Unlike the other ascertainment schemes explored in this study, this type of ascertainment demonstrates a bias only in the case of the (African$_x$, archaic; African$_y$, non-African) class of $f_4$-statistics (when only statistics with |Z| <15 SE on all sites were considered, S7 Table, S11 Fig). Another class of $f_4$-statistics is biased under this ascertainment scheme when all statistics are considered: $f_4$(non-African$_x$, archaic; African, non-African$_y$) (S11 Fig), and AFR MAF

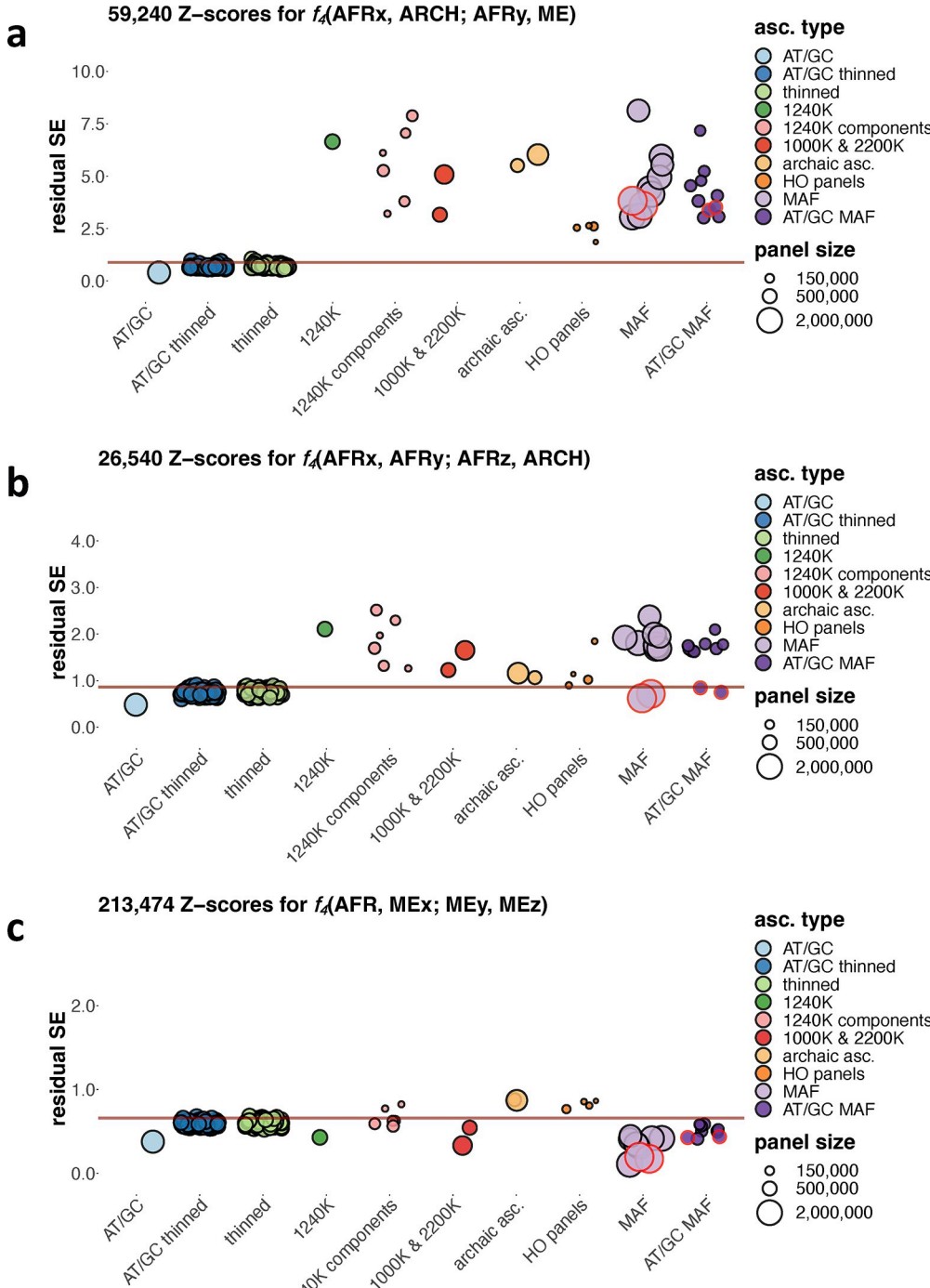

**Fig 5. Variance in $f_4$-statistic Z-scores resulting from ascertainment and random site subsampling expressed as residual standard deviation of a linear trend fitted to a scatterplot of Z-scores on unascertained vs. ascertained data (abbreviated as "residual SE" and expressed in the same units as $f_4$-statistic Z-scores).** Results are shown for three classes of $f_4$-statistics: $f_4$(African$_x$, archaic; African$_y$, Mediterranean/Middle Eastern), $f_4$(African$_x$, African$_y$; African$_z$, archaic), and $f_4$(African, Med/ME$_x$; Med/ME$_y$, Med/ME$_z$). The following abbreviations are used for naming the $f_4$-statistics classes: AFR, African populations; ARCH, archaic human individuals (Neanderthals and Denisovans); Med/ME or ME, Mediterranean and Middle Eastern populations. Results for ascertainment on variants common in Africans (either those having no detectable West Eurasian ancestry according to Fan *et al*. [67] or on all Africans in the SGDP dataset) are circled in red. Residual SE values for $f_4$-statistic Z-scores lying not far from 0 (absolute Z-scores on all sites < 15) are plotted. The 97.5% percentiles of all the thinned replicates combined, including those on all sites and

AT/GC sites, are marked by the brown lines. Size of the SNP panels is coded by point size, and the broad ascertainment types are coded by color according to the legend. Thirty eight ascertainment schemes were explored, identical to those in Fig 2.

ascertainment is unbiased in the case of the other 25 classes of $f_4$-statistics explored in this study (S7 Table, S11 Fig), which also translates into downstream analyses such as fits of admixture graph models (Figs 2 and S2, Tables 1 and S3–S5).

## Discussion

$f$-statistics [16] form a foundation for a range of methods ($qpWave$, $qpAdm$, $qpGraph$) that are used widely for studying population genetic history of humans and other species (see, for instance, [75,80–82]). Here, we focused on $f_4$-statistics, which are used as standalone tests for cladality [16,19] and underlie the $qpAdm$ method for fitting admixture models [27,28]. The other $f$-statistics ($f_2$ and $f_3$) can be defined as special cases of $f_4$-statistics [$f_2(A, B) = f_4(A, B; A, B)$ and $f_3(A; B, C) = f_4(A, B; A, C)$], and are subject to the same kinds of biases. The existence of bias in the case of non-outgroup ascertainment was recognized in a publication introducing a suite of methods relying on $f$-statistics [16], but its effects on large collections of $f_4$-statistics or on fits of diverse admixture graph models were not explored in that study and in subsequent studies. Since usage of ape, archaic or African genomes as outgroups is often unavoidable for

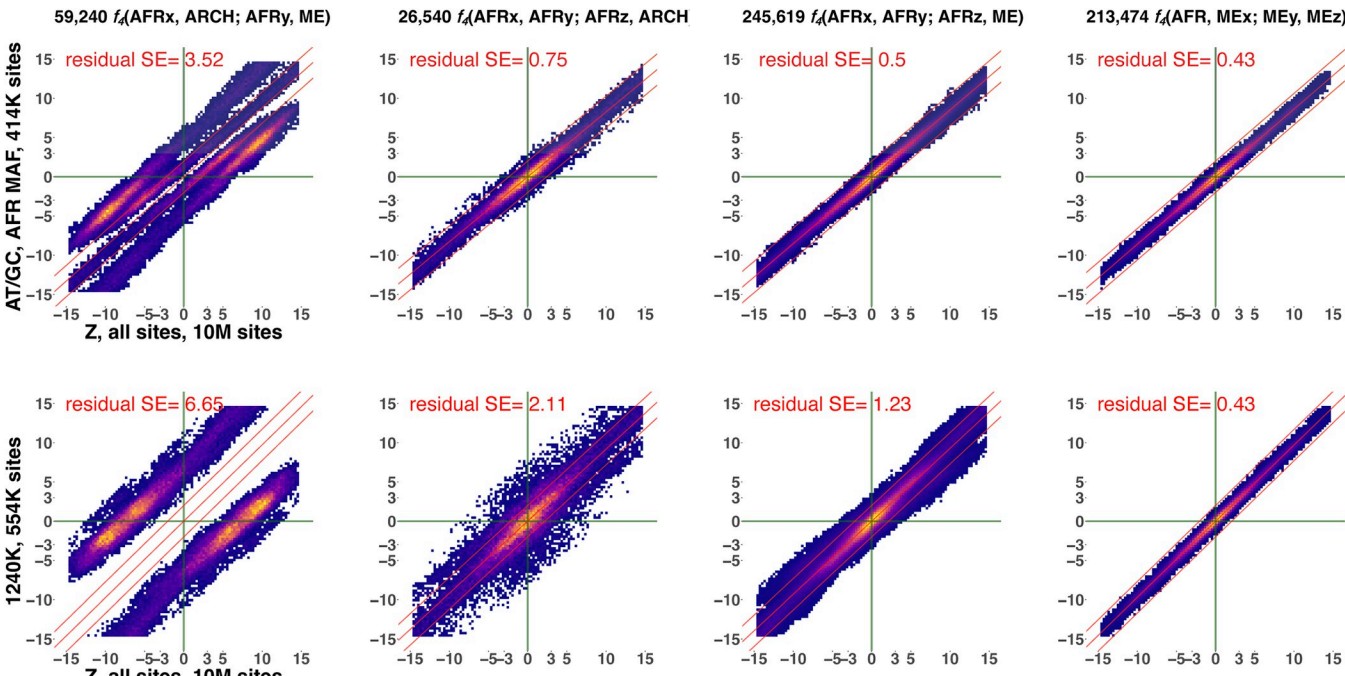

**Fig 6. Scatterplots illustrating the effects of two ascertainment schemes on Z-scores of $f_4$-statistics of four classes including African (abbreviated as AFR) and/or archaic (ARHC) and/or Mediterranean/Middle Eastern (ME) groups.** The $f_4$-statistic classes were selected to represent severe ascertainment bias (leftmost panels), moderate level of bias (two middle panels) and no bias (rightmost panels). The ascertainments selected are 1240K (a SNP enrichment panel most widely used in the archaeogenetic literature) and the new "African MAF" ascertainment scheme proposed in this study to mitigate bias for nearly all $f_4$-statistic classes. For results on other $f_4$-statistic classes see S13 Fig, and results for a wider range of ascertainment schemes are summarized in S11 and S12 Figs. Labels of $f_4$-statistic classes and numbers of statistics plotted are shown above the panels. Instead of individual points, heatmaps illustrating point density are shown. Z-scores on all sites (10 million sites, as indicated on the x-axes) are compared to Z-scores on ascertained datasets on the y-axes. Ascertainment schemes and site counts are shown on the y-axes. All plots include only statistics with absolute Z-scores below 15 on all sites. A linear model fitted to the data and lines representing ± 2 SE are shown in red. Residual standard deviations (residual SE) for those linear trends are shown in each plot in red.

calculation of $f_4$- and $D$-statistics and for construction of admixture graph or $qpAdm$ models (e.g., [4,6,9,12,56–58]), unbiased ascertainment on an outgroup that is not co-analyzed with other populations (as illustrated on simulated data in Fig 4A) is uncommon in practice. And frequently used SNP panels such as 1240K were built using very complex forms of non-outgroup ascertainment. Therefore, in this study we focused on practical rather than theoretical aspects of the SNP ascertainment bias problem and considered forms of non-outgroup ascertainment that are common in the literature on archaeogenetics of humans, including ascertainment on a phylogenetic outgroup co-analyzed with other populations.

The present analysis showed that $f_4$-statistics of specific types are affected by ascertainment bias. The most striking example we found is a class of statistics $f_4$(African$_x$, archaic; African$_y$, non-African). All statistics in this class are strongly biased in the same direction under the 1240K ascertainment (S14 Fig) and under all other non-random ascertainment schemes explored on real (S7 Table, S11D Fig) and simulated data (S6B Fig). In contrast, all $f_4$-statistic classes we explored including one or two African groups and non-Africans, or non-Africans only, turned out to be unbiased under the 1240K ascertainment (S7 Table, S11 Fig). Thus, numerous studies relying on fitting $qpAdm$ and/or admixture graph models including one African group and various non-Africans are probably minimally affected by ascertainment bias, as we also demonstrated on exhaustive collections of simple admixture graphs for few population sets (Figs 1 and 2, Tables 1 and S3–S5). When these classes of methods are applied to African population history, the situation is different, however. As we demonstrated, the 1240K panel emerges as biased when fits of simple admixture graphs including five African groups or one to two archaic and three to four African groups are considered (Figs 1 and 2, Tables 1 and S3–S5). We also demonstrated that the 1240K ascertainment affects fits of more complex admixture graphs including in all cases chimpanzee and Altai Neanderthal, and also four or six African groups and one or two groups with substantial non-African ancestry (S1 Text, S4 and S5 Figs). We expect fits of many other admixture graphs for Africans beyond those tested in this study to be affected by the 1240K ascertainment since the $f_4$-statistic classes $f_4$(African$_x$, archaic; African$_y$, non-African), $f_4$(African$_x$, archaic; African$_y$, chimpanzee), and $f_4$(African$_x$, African$_y$; African$_z$, X) are substantially biased under this ascertainment (S7 Table). These effects were reproduced on simulated data when true simulated graphs including "chimpanzee", one "archaic" lineage, and several "African" and "non-African" lineages were fitted to the data ascertained in various ways (Fig 3B).

In line with theoretical expectations, $f_4$-statistics including AMH groups only are largely unbiased under archaic ascertainment ([6,30], technical note published on the Daicel Arbor Biosciences product page). However, as compared to other SNP panels of similar size, archaic ascertainment increases variance in nearly all $f_4$-statistic classes of the types $f_4$(non-African$_x$, non-African$_y$; non-African$_z$, X) and $f_4$(non-African$_w$, non-African$_x$; non-African$_y$, non-African$_z$) (S7 Table, S11–S13 Figs). Increased variance in these cases can be explained by the low information content of an archaically ascertained panel: unlike the other non-random ascertainment schemes we tested, archaic ascertainment preserves most sites with nearly fixed ancestral variants and leads to just a moderate enrichment for common variants (DAF between 5% and 95%), especially if DAF is based on non-Africans (S3 Text, S6D Fig, S10 Table). Thus, the archaically ascertained panel includes a relatively small number of variants that are common in AMH and especially in non-Africans (S10 Table), and that increases the noise level. This elevated noise level in $f$-statistics under archaic ascertainment translates to reduced power to reject admixture graph models based on these $f$-statistics (Figs 1 and 2, Tables 1 and S3–S5). This effect was also reproduced on simulated data (S9 Fig). If archaic humans are included in an $f$-statistic or an admixture graph, archaic ascertainment is no longer guaranteed to be unbiased (see Fig 4A and 4B), and indeed due to the existence of the

Neanderthal to non-African gene flow it fails to fix the bias affecting the most problematic class of statistics $f_4$(African$_x$, archaic; African$_y$, non-African), as demonstrated on simulated data in S6B and S16 Figs.

Many ascertainment schemes such as the 1240K, 2000K, Illumina 650Y panels and MAF-based ascertainment on non-Africans skew average DAF across four populations in an $f_4$-statistic since these panels are enriched for derived variants common in non-Africans vs. Africans and in AMH vs. archaic humans (S3 Text, S10 Table). Overrepresentation of derived variants in certain groups of the quadruplet skews $f_4$-statistics. We conclude that two ascertainment schemes most often used for studies of African population history (1240K and archaic ascertainment) are not optimal for various reasons: overrepresentation of derived variants common in non-Africans in the former case and a small number of variants common in AMH in the latter case.

We found that there exists a non-outgroup ascertainment scheme that demonstrates a nearly optimal balance of bias and statistical power: restricting to variants that are common in a diverse collection of African groups. This scheme demonstrated a bias only in the case of the $f_4$(African$_x$, archaic; African$_y$, non-African) and $f_4$(non-African$_x$, archaic; African, non-African$_y$) classes of $f_4$-statistics among the 27 classes investigated (S7 Table, S11, S13, and S16 Figs). This scheme does not favor derived variants common in non-Africans and supplies many variants common in both Africans and non-Africans (S10 Table). While for many $f_4$-statitic classes and admixture graphs, the difference in performance of the AFR MAF and archaic ascertainment schemes is small (Table 1, S2, S3 and S11 Figs, S3–S5 and S7 Tables), the AFR MAF scheme is applicable when Neanderthals and Denisovans are co-analyzed (S2 and S3 Figs), while archaic ascertainment generates extreme shifts in $f_4$-statistics in this case (Figs 3B and S17; see also S4 Text). The AFR MAF scheme is also effective for analyses focused on non-Africans, demonstrating no elevated noise level typical for archaic ascertainment (S7 and S3 Tables). Thus, the AFR MAF ascertainment is the most widely applicable scheme among those explored in this study. According to our results on collections of admixture graphs (Table 1) and on $f_4$-statistic classes (S7 Table), a similar form of ascertainment, namely combining sites heterozygous in a single San and a single Yoruba individual (HO panels 4 & 5) is also largely unbiased, with the exception of statistics of the form $f_4$(African$_x$, archaic; African$_y$, non-African). However, this ascertainment is also noisier due to the low number of sites available. We tested several of the panels comprising the Affymetrix Human Origins SNP array (the largest of them), each ascertained as sites heterozygous in a high-coverage human genome from a selected population, since they were proposed to be "clean" forms of ascertainment in the publication where they were introduced [16]. HO panels are rarely used in practice individually because of their small size, which is especially problematic for ancient individuals with high rates of missing data, and our results confirm that this practice is justified.

As we demonstrated on simulated data, for a non-outgroup ascertainment to be unbiased it should be based on a population that is genetically close to the root (Fig 4B and 4C) (however, such an ascertainment usually has relatively low statistical power for rejection of incorrect admixture graph models, see S9 Fig). We note that in our analysis on simulated data the group where ascertainment was performed was co-modelled with the other groups, as is often done in practice (especially considering not only identical but also closely related groups). In the light of these results, archaic ascertainment's sub-optimal performance as a non-outgroup ascertainment is because Denisovans and Neanderthals have probably had a low long-term effective population size [34], and thus are highly drifted with respect to the root. Moreover, it is often unavoidable that individuals used for archaic ascertainment are used as sole representatives of the respective groups analyzed, and that is also problematic (Fig 4A). Africans, in contrast, have had much higher effective population sizes for a long period [34], and we

propose that restricting to variants common in a diverse set of African genomes is much more reliable (than archaic ascertainment or ascertainment on a single African population or individual) for preserving the spectrum of variants that existed at the root of archaic and anatomically modern humans (see Fig 4C). At the same time, AFR MAF ascertainment supplies enough variants that are common in non-Africans, making it also relatively powerful statistically for analyses focused on non-Africans.

An enrichment approach is powerful for large-scale ancient DNA research in Africa due to DNA preservation issues in the hot climate [6]. We did not test a range of allele frequency cutoffs or many alternative sets of individuals for AFR MAF ascertainment, and we do not propose a specific list of sites for a new DNA enrichment panel. However, our results imply that an effective approach for designing such a panel, which would also be useful for human archaeogenetic studies worldwide, would be to combine selection of the A/T and G/C mutation types with restricting to variants common in Africa. Frequencies of alleles at A/T and G/C loci are not affected by biased gene conversion (its rate depends on population heterozygosity [70]), these loci are not hypermutable, and are not affected by deamination damage in ancient DNA. As we demonstrated, restricting to A/T and G/C sites does not bias $f_4$-statistics (S7 and S10 Tables, S11 and S15 Figs) or admixture graph fits (Tables 1 and S3–S5). Another reason for taking A/T and G/C sites only is simply reducing the number of sites since enrichment reagents have limited capacity, and this ascertainment scheme with a 5% MAF threshold yields about 1.6 million variable sites on the "SGDP+archaic" dataset.

## Methods

### 1. Simulating genetic data

**1.1 Simulating the relationships of AMH and archaic humans with msprime v.0.7.4.**
Twenty-two chromosomes matching the size of the human chromosomes in the *hg19* assembly were simulated with a flat recombination rate (2 x $10^{-8}$ per nt per generation) and a flat mutation rate, 1.25 x $10^{-8}$ per nt per generation [83]. The standard coalescent simulation algorithm was used [84], and diploid genomes were assembled from these independently simulated 22 haploid chromosomes. Although this approach does not recapitulate the linkage disequilibrium pattern in real human genomes, it does not make a difference for simulating allele frequencies in deeply divergent groups since chromosome histories become quickly independent in the past [85].

The following groups were simulated: chimpanzee ("Chimp", one individual sampled at "present" in the simulation time), the Vindija Neanderthal ("Neanderthal", one individual sampled 2,000 generations or 50,000 years in the past, considering a generation time of 25 years), the high-coverage Denisovan "Denisova 3" ("Denisovan", one individual sampled 2,000 generations in the past), five African groups (10 individuals per group sampled at "present") and three non-African groups (10 individuals per group sampled at "present"). Five classes of simulated topologies are shown in S16B Fig; for a full list of simulation parameters and their values see S13 Table. Only one simulation iteration was performed for each combination of parameters.

We applied archaic ascertainment to the simulated data: restricting to sites polymorphic in the group composed of two "archaic" individuals, "Denisovan" and "Neanderthal" (this scheme reproduces the archaic ascertainment applied to real data, the "SGDP+archaic" dataset, in S16 and S17 Figs). For calculating *f*-statistics on unascertained and ascertained SNP sets, the software package *ADMIXTOOLS 2* [23] was used. Since there was no missing data at the group level and all individuals were diploid, we first calculated all possible $f_2$-statistics for 4 Mbp-sized genome blocks (with the "*maxmiss = 0*", "*adjust_pseudohaploid = FALSE*", and

"*minac2 = FALSE*" settings), and then used them for calculating $f_4$-statistics as linear combinations of $f_2$-statistics. This protocol was used for generating the results shown in S4 Text and S16C–S17C Figs.

**1.2 Simulating the relationships of AMH and archaic humans with msprime v.1.1.1.**
More realistic simulations were performed with *msprime v.1.1.1* which allows accurate simulation of recombination and of multi-chromosome diploid genomes relying on the Wright-Fisher model [76,85]. We simulated three chromosomes (each 100 Mb long) in a diploid genome by specifying a flat recombination rate ($2 \times 10^{-8}$ per nt per generation) along the chromosome and a much higher rate at the chromosome boundaries ($\log_e 2$ or ~0.693 per nt per generation, see https://tskit.dev/msprime/docs/stable/ancestry.html#multiple-chromosomes). A flat mutation rate, $1.25 \times 10^{-8}$ per nt per generation [83], and the binary mutation model were used. To maintain the correct correlation between chromosomes, the discrete time Wright-Fischer model was used for 25 generations into the past, and deeper in the past the standard coalescent simulation algorithm was used (as recommended by Nelson *et al.* [85]).

The following groups were simulated: chimpanzee ("chimp", one individual sampled at "present" in the simulation time), the Altai Neanderthal ("Neanderthal 1", one individual sampled 3,790 generations in the past), the Vindija Neanderthal ("Neanderthal 2", one individual sampled 1,700 generations in the past), the high-coverage Denisovan "Denisova 3" ("Denisovan", one individual sampled 1,700 generations in the past), two African groups ("African 1" and "African 2", 10 individuals per group sampled at "present") and two non-African groups ("non-African 1" and "non-African 2", 10 individuals per group sampled at "present"). The topology is shown in Fig 3A; for a full list of simulation parameters see S13 Table. Ten simulation iterations were performed for each combination of parameters, and two combinations were tested: with or without the Neanderthal to non-African gene flow.

Upon assessing genetic distances across the simulated groups using $F_{ST}$, the following ascertainment schemes were applied:

1. "HO, 1 panel", that is restricting to sites that are heterozygous in a randomly selected individual from the "African 2" group (this scheme simulates the generation of one Human Origins SNP panel [16]);

2. "HO, 4 panels", taking heterozygous sites from one randomly selected individual per "AMH" population ("African 1", "African 2", "non-African 1", "non-African 2") and merging these SNP sets (this scheme simulates the generation of the whole Human Origins SNP array [16]);

3. "AFR MAF", restricting to sites having high minor allele frequency (>5%) in the union of "African" groups "African 1" and "African 2";

4. "non-AFR MAF", restricting to sites having high minor allele frequency (>5%) in the union of "non-African" groups "non-African 1" and "non-African 2" (this scheme simulates MAF ascertainment on a non-African continental meta-population);

5. "global MAF", restricting to sites having high minor allele frequency (>5%) in the union of all "AMH" groups "African 1", "African 2", "non-African 1", and "non-African 2";

6. archaic ascertainment, restricting to sites polymorphic in the group composed of three "archaic" individuals, "Denisovan", "Neanderthal 1", and "Neanderthal 2".

**1.3 Simulating random admixture graphs and simple trees with msprime v.1.1.1.**
Genetic histories in the form of random admixture graphs were simulated using the *msprime*

*v.1.1.1* settings described above. We simulated histories of four complexity classes: including 8 or 9 sampled non-outgroup populations, one outgroup, and 4 or 5 pulse-like admixture events. Demographic events were separated by date intervals ranging randomly between 1,500 and 8,000 generations, with an upper bound on the tree depth at 40,000 generations. To be more precise, demographic events were not placed in time entirely randomly, but were tied to one or few other events of the same "topological depth" within the graph, as illustrated by examples of the simulated topologies in S7 Fig. The same principle was applied to the sampling dates, which were tied to other demographic events such as divergence and admixture of other populations. The random graph topologies and parameter sets were generated using the *random_-sim* function from the *ADMIXTOOLS 2* package: https://uqrmaie1.github.io/admixtools/reference/random_sim.html

Outgroups facilitate automated exploration of graph topology space. The outgroup branches diverged from the other populations at 40,000 generations in the past and had a large constant effective population size of 100,000 diploid individuals. Other effective population sizes were constant along each edge and were picked randomly from the range of 2,000–40,000 diploid individuals. Admixture proportions for all admixture events varied randomly between 10% and 40%. The root of the simulation and the root of all non-outgroup populations were sampled, and the other branches were sampled at tips exclusively. This setup generates groups sampled at widely different dates in the past (from 0 to ca. 40,000 generations) or, in other words, located at various genetic distances from the root (Fig 4B). The outgroup population was sampled at the "present" of the simulation. Sample sizes for all populations were identical: 10 diploid individuals with no missing data.

For subsequent analyses we selected only simulations where pairwise $F_{ST}$ for groups were in the range characteristic for anatomically modern and archaic humans (in each simulation there was at least one $F_{ST}$ value below 0.15; see S7 Fig). In this way, 20 random topologies were simulated per complexity class. Each topology was simulated only once, and 80 simulations were generated in total (see examples of the topologies and respective $F_{ST}$ distributions in S7 Fig). Another set of simulations was prepared with the same topologies and parameters, except for the effective population size on the outgroup branch which was set at 1,000 diploid individuals instead of 100,000.

The following ascertainment schemes were applied to the outcomes of these randomized simulations: 1) HO one-panel ascertainment (repeated for all simulated groups including the outgroup and root groups, generating 920 ascertained datasets); 2) HO four-panel ascertainment (10 random sets of four groups excluding the outgroup and root groups were explored per topology, generating 800 ascertained datasets); 3) ascertainment on sites polymorphic in a group composed of three randomly selected individuals, with only one individual per group considered (10 random sets of three groups excluding the outgroup and root groups were explored per topology, generating 800 ascertained datasets); and 4) MAF ascertainment, that is restricting to sites having MAF >5% in random meta-groups (10 random sets of four groups excluding the outgroup and root groups were explored per topology, generating 800 ascertained datasets). Group sets used for each ascertainment were recorded. Genetic distances ($F_{ST}$) were calculated for all populations (including the outgroup and the last common ancestor of all non-outgroup populations) vs. the root sample (Fig 4B).

Alternatively, simple trees were simulated using the *msprime v.1.1.1* settings described above. A tree of four groups conforming to the $f_4$-statistic (*A, B; C, O*) was simulated using *msprime v.1.1.1*, with a tree depth of 4,000 generations (S10 Fig). All the groups had a uniform effective population size of 100,000 diploid individuals, except for a bottleneck happening immediately after the *A-B* divergence (at 1,999 generations in the past) and lasting until "present" in the simulation time. The following bottleneck classes were simulated: no bottleneck

(control), 10x, 100x, 1,000x, and 10,000x reduction in effective population size. For each bottleneck class, 20 independent simulations were performed. All the samples were drawn at "present": sample sizes were 25, 25, 25 and 10 for populations *A*, *B*, *C* and *O*, respectively (except for the 10,000x bottleneck class since group *A* included 10 individuals only in that case). Three ascertainment schemes were tested for the simulated trees: 1) HO one-panel ascertainment (repeated for all simulated groups, including group *O*); 2) restricting to sites having MAF >5% (or 10%, or 2.5%) in the union of groups A and B composed of 50 diploid individuals; and 3) removal of sites with *derived* allele frequency >95% (or 90%, or 97.5%) in the union of groups A and B. The latter ascertainment scheme was added since the ascertainments we tested on real data deplete the derived end of the allele frequency spectrum more than the ancestral end (S10–S12 Tables).

For calculating *f*-statistics and fitting admixture graphs to unascertained and ascertained SNP sets, the *ADMIXTOOLS 2* [23] software package was used. Since there was no missing data and all individuals were diploid, we first calculated all possible $f_2$-statistics for 4 Mbp-sized genome blocks (with the "*maxmiss = 0*", "*adjust_pseudohaploid = FALSE*", and "*minac2 = FALSE*" settings) and then used them for calculating $f_4$-statistics as linear combinations of $f_2$-statistics or for fitting admixture graphs (with the "*numstart = 100*" and "*diag = 0.0001*" settings). This calculation protocol was used for generating the results shown in Figs 3 and S6–S10. When true admixture graphs were fitted to ascertained data, full population samples of 10 individuals were used by default, and in some cases, as indicated in the figure legends, the individual used for ascertainment was used as the only representative of the respective population. Unless stated otherwise, phylogenetic outgroups were included in fitted graphs; in other words, they were co-analyzed with the other groups.

For assessing the power of ascertained simulated datasets to reject incorrect admixture graph models, we first generated a set of such incorrect graphs per each simulated topology. For that purpose, an algorithm for finding well-fitting topologies (*findGraphs* from the *ADMIXTOOLS 2* package) was started on unascertained data 300 times, seeded by random graphs containing either the simulated number of admixture events (*n*, 100 runs), or *n-1* events (100 runs), or *n+1* events (100 runs). For a list of settings for the *findGraphs* algorithm see Maier *et al.* [23]. Thousands of diverse graphs explored by *findGraphs* in the process of topology optimization were generated in this way for each simulated graph, and 100 poorly-fitting graphs were randomly picked from a subset of these graphs having LL scores between 70 and 300. This subset of graphs was then fitted to all ascertained datasets derived from the same simulated admixture graph, and the analysis was repeated for all simulated topologies.

## 2. Constructing the set of real data

We used the *cteam-lite* dataset described Mallick *et al.* [36], composed of the full SGDP set (300 high-coverage genomes from present-day populations), the chimpanzee genome (pseudo-haploid genotype calls, see http://hgdownload.cse.ucsc.edu/goldenPath/panTro2/bigZips/), and the Altai Neanderthal, "Denisova 3" Denisovan, Ust'-Ishim, WHG Loschbour, and LBK Stuttgart ancient genomes (see SI section 3 in Mallick *et al.* [36]). We supplemented *cteam-lite* by 44 present-day African genomes sequenced using the SGDP protocols by Fan *et al.* [67], the Vindija Neanderthal's genome [33], and the genome of an ancient African forager individual I10871 sequenced by Lipson *et al.* [9] (S1 Table). Sites polymorphic in this set of 352 individuals were extracted from the *cteam-lite* files of the "hetfa" format using the *cpoly* tool [36]: alleles were grouped into derived and ancestral (polarized) according to the chimpanzee genome; missing data and heterozygous sites were allowed. For each genome, we used individual base quality masks included in *cteam-lite* or constructed using the same protocol

for other genomes (Vindija Neanderthal and those sequenced by Fan *et al.* [67]): minimum base quality was set by default at 1, as recommended by Mallick *et al.* ([36], see SI section 3 in that study), which discarded lowest-quality regions marked as "0", "?", or "N". The individual I10871 was not included in most analyses in this study (except for the complex admixture graphs in S1 Text, and S4 and S5 Figs) due to its relatively high rate of deamination errors.

The resulting dataset prior to missing data removal and ascertainment includes 94,691,841 biallelic autosomal SNPs (S2 Table). To keep the polarity of alleles, all data manipulations and ascertainments were performed using *PLINK v.2.0 alpha* [86]. For calculating $f_4$-statistics, sets of continental-level meta-populations were selected (e.g., Africans and East Asians or Africans and archaic humans) and then $f_4$-statistics were calculated for all possible combinations of populations in the resulting subset of the "SGDP+archaic" dataset, with no missing data (at the population level) allowed *within the selected subset*. This was done to avoid potential biases associated with data missing non-randomly across groups. Alternatively, $f_4$-statistics were drawn randomly from a certain class of statistics, and no missing data (at the population level) were allowed in the resulting *population quadruplets*.

## 3. Influence of ascertainment on fits of admixture graphs to real data

First, we fitted all possible graphs including two admixture events (32,745 distinct topologies with no fixed outgroup) for three combinations of groups: 1) one archaic individual, three African groups, and one African group with substantial West Eurasian-related ancestry (Altai Neanderthal, Ju|'hoan North, Biaka, Yoruba, and Agaw, respectively); 2) five deeply-divergent ancient and present-day non-African groups (Ust'-Ishim, Papuan, Onge, LBK Stuttgart, Even); and 3) five deeply-divergent present-day non-African groups (Papuan, Onge, Palestinian, Even, Mala). These three sets of simple graphs were fitted to all sites, AT/GC sites, and 1240K sites (no missing data were allowed at the group level within these sets of five populations); 5,000 best-fitting models were selected according to LL scores on all sites and WRs of those models were compared across SNP sets (Fig 1).

Next, we explored the same exhaustive set of admixture graph topologies including five groups and two admixture events on the wider collection of ascertainment schemes. Twelve combinations of five groups including up to two archaic humans, up to five African groups, and up to five non-African groups were tested. To ensure fair comparison across at least a subset of population combinations, as a starting point for generating ascertained site sets we used either 11,706,773 sites (with no missing data at the group level) polymorphic in a set of 48 archaic and African groups composed of 97 individuals in total; or 10,051,585 such sites in a set of 59 archaic, African, European, and Middle Eastern groups composed of 120 individuals in total; or 5,296,653 such sites in a set of 51 Papuan, Native American/Siberian, European, Anatolian, and Caucasian groups composed of 112 individuals in total (S1 and S2 Tables).

We examined fits of these collections of admixture graphs to the real data from different perspectives. (1) We considered just 5,000 topologies per each population quintuplet that are best-fitting on the unascertained site set (Figs 1, S1 and S2) or all 32,745 topologies tested (S3 Fig). (2) We also considered alternative admixture graph fit metrics, LL or WR. LL as a fit metric (see left-hand panels in S2 and S3 Figs) is more accurate than WR, but difficult to compare across different population sets. Finally, in addition to (squared) Pearson correlation coefficient of admixture graph fits on ascertained vs. unascertained data (Figs 1, 2 and S1–S3) as a way of measuring ascertainment bias, we considered the fraction of all possible admixture graph models of a certain complexity that are fitting the data poorly ("rejected") under ascertainment (WR >3 SE) but fitting well ("accepted") on all sites (WR <3 SE). The fraction of all possible models that are fitting the data well under ascertainment (WR <3 SE) but fitting

poorly on all sites (WR >3 SE) was used to measure the power to reject incorrect admixture graph topologies.

## 4. Automated inference of fitting admixture graphs on real data

The 12-population admixture graph published by Lipson *et al.* [9] (and later used as a skeleton graph in Lipson *et al.* [12]) and simpler 7- and 10-population intermediate graphs presented in the former study were revisited by Maier *et al.* [23], and thousands of alternative well-fitting graphs of the same complexity were found using the *find_graphs* function from the *ADMIX-TOOLS 2* package (https://uqrmaie1.github.io/admixtools/articles/graphs.html). Maier *et al.* [23] used the 1240K dataset only, and in the current study we re-fitted the admixture graphs found by the algorithm on the 1240K SNP panel to the AT/GC and unascertained datasets derived from the "SGDP+archaic" dataset, and also repeated automated admixture graph inference on these two additional SNP sets. Advantages and pitfalls of automated admixture graph inference are described in detail in Maier *et al.* [23], along with justifications for the specific protocol used in that study, and here we used the protocols identical to those employed by Maier *et al.* [23]. We first calculated all possible $f_2$-statistics for 4 Mbp-sized genome blocks (with the "*maxmiss = 0*", "*adjust_pseudohaploid = FALSE*", and "*minac2 = 2*" settings, see Maier *et al.* [23] for details on the settings) and then used them for fitting admixture graphs (with the "*numstart = 100*" and "*diag = 0.0001*" settings) and for automated admixture graph inference with the *find_graphs* function (see the Methods section in Maier *et al.* [23] for a complete list of arguments for this function). Only one topology constraint was used at the graph space exploration step: chimpanzee was assigned as an outgroup.

## Supporting information

**S1 Fig. Two alternative approaches for measuring the effect of ascertainment bias on admixture graph fits are illustrated using one population combination, "Denisovan, Khomani San, Mbuti, Dinka, Mursi".** 1) residual standard deviation (residual SE) of linear trends and 2) squared Pearson correlation coefficient ($R^2$) for two admixture graph fit metrics (worst $f_4$-statistic residuals, WR, or log-likelihood scores, LL) calculated on unascertained vs. ascertained data. Five thousand best-fitting graphs (according to LL on all sites) of 32,745 possible graphs were selected, and correlation of LL (left-hand panels) or WR (right-hand panels) was explored for graphs fitted on all sites and on ascertained datasets. Results for ascertainment on variants common in Africans (either those having no detectable West Eurasian ancestry or all Africans in the SGDP dataset) are circled in red. As a starting point for generating different ascertained datasets, we used 11,706,773 sites (with no missing data at the group level) polymorphic in a set of 48 archaic and African groups composed of 97 individuals (S1 Table). Thirty-eight site subsampling schemes were explored (see a list in the legend for Fig 2). The size of the resulting SNP panels is coded by point size, and ten broad ascertainment types are coded by color according to the legend in the upper right corner. The 97.5[th] (in the case of residual SE) or 2.5[th] (in the case of $R^2$) LL or WR percentiles of all the thinned replicates combined, including those on all sites and AT/GC sites, are marked by brown lines. Areas of the plots where ascertainments are considered biased according to these thresholds are highlighted in red on the left-hand side of the plots. Scatterplots illustrating effects of selected ascertainment schemes (marked with numbers 1 to 7) on LL or WR are shown in the middle of the figure. Each dot on these scatterplots corresponds to a distinct admixture graph topology. (PDF)

**S2 Fig. Variance in fits of a collection of simple admixture graphs (five groups and two admixture events) resulting from ascertainment or random site subsampling expressed as squared Pearson correlation coefficient ($R^2$).** Five thousand best-fitting graphs (according to log-likelihood scores on all sites) of 32,745 graphs were selected for each combination of populations, and correlation of admixture graph log-likelihood scores (LL) or worst $f_4$-statistic residuals (WR) was explored for graphs fitted to unascertained vs. ascertained datasets. Results are shown for twelve population combinations indicated in plot titles (panels **a, b**). Results for ascertainment on variants common in Africans (either those having no detectable West Eurasian ancestry or on all Africans in the SGDP dataset) are circled in red. As a starting point for generating different ascertainments, we used either 11,706,773 sites (with no missing data at the group level) polymorphic in a set of 48 archaic and African groups composed of 97 individuals in total, or 10,051,585 such sites in 59 archaic, African, European, and Middle Eastern groups composed of 120 individuals, or 5,296,653 such sites in 51 Papuan, Native American/Siberian, European, Anatolian, and Caucasian groups composed of 112 individuals (S1 Table). Thirty eight site subsampling schemes were explored (see a list in the legend for Fig 2). The size of the resulting SNP panels is coded by point size, and ten broad ascertainment types are coded by color according to the legends. $R^2$ values for LL are plotted in the left-hand panels, and $R^2$ values for WR are plotted in the right-hand panels. The 2.5th LL or WR percentiles of all the thinned replicates combined, including those on all sites and AT/GC sites, are marked by brown lines.
(PDF)

**S3 Fig. Variance in fits of an exhaustive collection of 32,745 simple admixture graphs (five groups and two admixture events) resulting from ascertainment or random site subsampling expressed as squared Pearson correlation coefficient ($R^2$).** Correlation of admixture graph log-likelihood scores (LL) or worst $f_4$-statistic residuals (WR) was explored for graphs fitted to unascertained vs. ascertained datasets. Results are shown for twelve population combinations indicated in plot titles (panels **a, b**). Results for ascertainment on variants common in Africans (either those having no detectable West Eurasian ancestry or on all Africans in the SGDP dataset) are circled in red. As a starting point for generating different ascertainments, we used either 11,706,773 sites (with no missing data at the group level) polymorphic in a set of 48 archaic and African groups composed of 97 individuals, or 10,051,585 such sites in 59 archaic, African, European, and Middle Eastern groups composed of 120 individuals, or 5,296,653 such sites in 51 Papuan, Native American/Siberian, European, Anatolian, and Caucasian groups composed of 112 individuals (S1 Table). Thirty eight site subsampling schemes were explored (see a list in the legend for Fig 2). The size of the resulting SNP panels is coded by point size, and ten broad ascertainment types are coded by color according to the legends. $R^2$ values for LL are plotted in the left-hand panels, and $R^2$ values for WR are plotted in the right-hand panels. The 2.5th LL or WR percentiles of all the thinned replicates combined, including those on all sites and AT/GC sites, are marked by brown lines.
(PDF)

**S4 Fig. Results of a search for optimal admixture graph models based on population sets and admixture graph complexity classes from Lipson *et al.* [9].** The search for optimal topologies was performed on three datasets: 1240K, AT/GC mutation types, and all sites. (**a**) The published 10-population model [9] with 8 admixture events is plotted, with parameter estimates obtained on the three datasets, and with corresponding worst $f_4$-statistic residuals (WR) and log-likelihood scores (LL) shown above the graphs. Distinct populations are colored along with their ancestral lineages; for instance, the cluster of West African populations is colored in brown. (**b**) Density plots illustrating fits of ca. 10,000 distinct topologies per dataset (found

with *findGraphs* [23]) in the LL vs. WR coordinates. Green vertical lines mark median LL of the highest-ranking newly found model fitted to bootstrap replicates of the dataset, and red lines mark 95th percentile of that distribution. Position of the published 10-population model in these coordinates is marked by the cyan dot. (**c**) Highest-ranking models (according to LL) with 8 admixture events found with *findGraphs* on each dataset. Populations and ancestral lineages are color-coded in the same way as in panel **a**. WR and LL of these models are shown above the plots. We also compared the fit (i.e., LL) of the highest-ranking newly found model with that of the published model on each dataset relying on a bootstrap resampling approach: comparison of two LL distributions on 500 resampled sets of SNP blocks [23]. For all three SNP sets, the difference in LL between the highest-ranking model found by the automated search and the published model was statistically significant, with empirical two-tailed p-values ranging from <0.002 to 0.032. In other words, it was shown for the three datasets explored that the published model fits significantly worse than the newly found highest-ranking models (however, this does not prove that the newly found models approximate the true population history better, see an analysis in Fig 1 from Maier *et al.* [23]).
(PDF)

**S5 Fig. Density scatterplots illustrating the effects of the 1240K and AT/GC ascertainments on log-likelihood scores, LL (a, c), or worst $f_4$-statistic residuals, WR (b, d), of admixture graphs found using *findGraphs* on the set of groups from Lipson *et al.* [9].** (**a, b**) Between 9,927 and 9,990 unique topologies including 10 groups and 8 admixture events were outcomes of 10,000 independent runs of the *findGraphs* algorithm on three datasets (shown in three columns). (**c, d**) Between 779 and 971 unique topologies including 7 groups and 4 admixture events were outcomes of 2,000 independent runs of the *findGraphs* algorithm on the three datasets. On x-axes LL or WR of admixture graphs fitted to all sites are shown. LL or WR of admixture graphs fitted to the 1240K SNP set are shown in the upper row on the y-axes, and LL or WR of admixture graphs fitted to AT/GC sites are shown in the lower row. Pearson correlation coefficients for these two sets of admixture graph fit metrics are displayed beside each plot in red. The WR threshold used often for fitting models (3 SE) is marked with green vertical and horizontal lines.
(PDF)

**S6 Fig. Effects of ascertainment on $f_4$-statistics explored on the simulated demographic history chosen as a case study.** (**a**) Influence of ascertainment on $F_{ST}$ across all population pairs on data simulated with or without the "Neanderthal" admixture in "non-Africans". Median $F_{ST}$ values are shown across 10 simulation iterations. Results for seven types of SNP sets are presented: 1) unascertained sites (on average 5.55M polymorphic sites without missing data); 2) HO one-panel ascertainment, based on the "African 2" group (500K sites on average across simulation iterations); 3) HO four-panel ascertainment (on the "African 1", "African 2", "non-African 1", and "non-African 2" groups, 1.34M sites on average); 4) archaic ascertainment (1.05M sites on average); 5) AFR MAF ascertainment, that is restricting to sites with MAF >5% in the union of "African 1" and "African 2" groups (1.66M sites on average); 6) global MAF ascertainment on the union of "African 1", "African 2", "non-African 1", "non-African 2" (2.75M sites on average); 7) non-African MAF ascertainment (1.04M sites on average). (**b**) Boxplots summarizing various $f_4$-statistics of the form $f_4$("African 1", "archaic"; "African 2", "non-African") for the two simulated topologies, on unascertained and ascertained data across 10 simulation runs. $f_4$-statistics are shown on the left and their Z-scores are shown on the right. (**c**) Boxplots summarizing various $f_4$-statistics of the form $f_4$("non-African 1 or 2", "archaic"; "African 1", "non-African 2 or 1") for the two simulated topologies, on unascertained and ascertained data across 10 simulation runs. $f_4$-statistics are shown on the left and

their Z-scores are shown on the right. (**d**) Simplified derived allele frequency (DAF) spectra for the unascertained and ascertained datasets. DAF was defined on the union of groups "non-African 1" and "non-African 2", and three allele frequency bins were defined: $<5\%$, $> = 5\%$ & $< = 95\%$, $>95\%$. Median site counts across 10 simulation iterations are presented. Similar results on real data are shown in S10 Table. (**e** and **f**) Boxplots summarizing DAF-stratified $f_4$-statistics of the form $f_4$("African 1", "archaic"; "African 2", "non-African 1 or 2") (**e**) and their Z-scores (**f**) for the two simulated topologies, on unascertained and ascertained data. In all $f$-statistic notations, the simulated populations are abbreviated as follows: a1 and a2, "Africans" 1 and 2; c, "chimpanzee"; d, "Denisovan"; n1 and n2, "Neandertals" 1 and 2; na1 and na2, "non-Africans" 1 and 2.
(PDF)

**S7 Fig. Examples of simulated genetic histories in the form of random admixture graphs including 8 (a, b) or 9 (c, d) non-outgroup populations and 4 (a, c) or 5 (b, d) admixture events.** One topology per graph complexity class is shown as an example, along with $F_{ST}$ values for all population pairs. Time in generations is shown on the y-axis (for visual clarity, points on this axis are not spaced proportionately). Effective population sizes (in diploid individuals) are shown beside each edge. Sampled populations are labelled by letters in squares. Outgroups are not visualized; their divergence was placed at 40,000 generations ago for all the simulated histories, and their effective population size was 100,000 or alternatively 1,000 diploid individuals (both series of simulations were prepared). The color gradients have no special meaning and are used for visual clarity only.
(PDF)

**S8 Fig. Derived allele frequency (DAF) spectra (derived allele count in a sample of 20 chromosomes vs. proportion of sites) in simulated root, outgroup, and non-outgroup populations grouped according to the level of ascertainment bias.** The spectra shown here are based only on sites polymorphic in a sample of 20 chromosomes drawn at the root of the simulation. We note that outgroups had an effective population size of 100,000 diploid individuals and were included in the fitted admixture graphs; in other words, they were co-analyzed with the other populations. Populations sampled at branch tips ("non-OG groups") are binned by worst $f_4$-statistic residual (WR) of the true graph under HO one-panel ascertainment based on that population: 0 to 4, 4 to 10, and 10 to 100 SE. The boxplots summarize DAF across all the simulated admixture graph topologies. DAF bins are shown in three separate panels with different y-axis ranges: 0 derived alleles; 1 to 9 and 20 derived alleles; 10 to 19 derived alleles.
(PDF)

**S9 Fig. Assessing the power of ascertained SNP datasets to reject incorrect admixture graphs.** A set of 100 topologically diverse poorly-fitting graphs was generated for each simulated random topology (see Methods) and fitted to both non-ascertained data and to all SNP sets ascertained using the HO one-panel scheme. (**a**) Boxplots summarizing distributions of worst $f_4$-statistic residuals (WR) of incorrect graphs fitted to datasets of several types: non-ascertained, ascertained on the outgroup with an effective population size of 100,000 that was co-modelled with the other populations ("OG"), on the root population sample that was not co-modelled with the other populations ("true root"), on the root of all non-outgroup populations that was not co-modelled with the other populations ("non-OG root"), and on non-outgroup populations co-modelled with the other populations ("non-OG leaves"). Results for all the simulated topologies were pooled. (**b**) SNP sets generated by HO one-panel ascertainment on simulated data in the "bias" vs. "power" coordinates. WR of the true simulated graph serves as a measure of ascertainment bias (on the x-axis), and median difference in WRs of 100

poorly-fitting graphs fitted to ascertained and non-ascertained data (on the y-axis) serves as a measure of power to reject incorrect models. Results are shown separately for SNP sets ascertained on OG, true root, non-OG root, and non-OG groups. Each dot represents an ascertained SNP set. (**c**) This series of plots is similar to that in panel **b**, but another measure of statistical power is used: the proportion of graphs which fit non-ascertained data poorly (WR >3 SE) but fit ascertained data well (WR <3 SE).
(PDF)

**S10 Fig. The influence of ascertainment on $f_4$-statistic cladality tests in the case of a simple tree $(O, (C, (A, B)))$ shown in (a).** Effective population size was constant across the tree, except for a drop in group $A$'s size at 1,999 generations in the past: from 100,000 to 10,000, 1,000, 100, or 10 diploid individuals. There was also a set of control simulations without any reduction in effective population size. In panel (**b**), $f_4$-statistics $f_4(A, B; C, O)$ and their Z-scores on 20 non-ascertained simulated datasets per bottleneck class and on datasets ascertained in various ways are summarized with boxplots. The ascertainment schemes are as follows: 1) keeping sites with MAF >5% in the union of groups $A$ and $B$ (abbreviated as "A+B MAF"); 2) keeping sites with *derived* allele frequency <95% in the union of groups $A$ and $B$ (abbreviated as "A+B DAF"); 3) HO one-panel ascertainment on either group $A$, $B$, $C$, or $O$ (results pooled for all ascertainment groups are shown). Very similar results were obtained for other MAF (2.5%, 10%) and DAF thresholds (90%, 97.5%) used for ascertainment; these results are not shown for brevity. In the case of HO ascertainment, no results are shown for simulations with the smallest effective size of group $A$ since too few SNPs were available due to rapid fixation of variants in the population with the extremely low effective size.
(PDF)

**S11 Fig. Variance in $f_4$-statistic Z-scores resulting from ascertainment and random site subsampling expressed as residual standard deviation of a linear trend fitted to a scatterplot of Z-scores on unascertained vs. ascertained data (abbreviated as "residual SE" and expressed in the same units as $f_4$-statistic Z-scores).** Results for ascertainment on variants common in Africans (either those having no detectable West Eurasian ancestry according to Fan *et al.* [67] or on all Africans in the SGDP dataset) are circled in red. Results are shown for 27 classes of $f_4$-statistics indicated in plot titles in panels **a**-**e**. The following abbreviations are used for naming the $f_4$-statistics classes: AFR, African populations; AMRSIB, Native American and Siberian populations; ARCH, archaic human individuals (Neanderthals and Denisovans); chimp, chimpanzee; EAS, East Asian populations; EUR, European populations; ME, Mediterranean and Middle Eastern populations; PAP, Papuan and Australian populations. Residual SE values for $f_4$-statistic Z-scores lying not far from 0 (absolute Z-scores on all sites < 15) are plotted in the right-hand panels, and residual SE values for all Z-scores are plotted in the left-hand panels. The 97.5% percentiles of all the thinned replicates combined, including those on all sites and AT/GC sites, are marked by the brown lines. Most y-axis scales are the same (from 0 to 2.5 SE), except for few cases in panels **a**, **b**, **c**, and **d**. Size of the resulting SNP panels is coded by point size, and ten broad ascertainment types are coded by color according to the legends. Thirty eight site subsampling schemes were explored (see a list in the legend for Fig 2).
(PDF)

**S12 Fig. Violin plots illustrating the effects of 10 ascertainment schemes on Z-scores of 14 $f_4$-statistic classes that were explored exhaustively: statistics including $x$ African, $y$ archaic, and $z$ Mediterranean/Middle Eastern groups (abbreviated as ME).** For brevity, $f_4$-statistic classes are labelled on y-axes as $x,y,z$. For instance, "2.1.1" stands for all possible $f_4$-statistics including two African, one archaic, and one Mediterranean/Middle Eastern groups. Z-score

difference was calculated as the Z-score on all sites (ca. 10 million sites) minus the Z-score on an ascertained dataset, and the distributions of absolute difference values are visualized. Ascertainment types and site counts are shown in plot titles. Only statistics with absolute Z-scores below 15 on all sites were considered for this analysis.
(PDF)

**S13 Fig. Scatterplots illustrating the effects of four ascertainment schemes on Z-scores of $f_4$-statistics of 14 classes including African (abbreviated as AFR) and/or archaic (ARCH) and/or Mediterranean/Middle Eastern groups (ME).** The class labels and numbers of statistics plotted are shown in the top row of each panel (**a**, **b**, and **c**). Instead of individual points, heatmaps illustrating point density are shown. Z-scores on all sites (ca. 10 million sites, as indicated on the x-axes) are compared to Z-scores on ascertained datasets on the y-axes. Ascertainment types and site counts are shown on the y-axes. All plots are based only on statistics with absolute Z-scores below 15 on all sites. A linear trend fitted to the data and lines representing ± 2 SE are shown in red. Residual SE values of those linear trends are shown in each plot in red.
(PDF)

**S14 Fig. A scatterplot illustrating the effect of the 1240K ascertainment on Z-scores of $f_4$-statistics $f_4$(African$_x$, archaic; African$_y$, non-African), where any non-African groups except for the Mediterranean and Middle Eastern groups were considered (results for the latter groups are shown in S13 Fig).** The following abbreviations are used: AFR, African populations; ARCH, archaic human individuals (Neanderthals and Denisovans); non-AFR, non-African populations. Instead of individual points, a heatmap illustrating point density is shown. Z-scores on all sites are compared to Z-scores on the 1240K dataset on the y-axis (site counts varied across population quadruplets since no missing data at the group level was allowed in each quadruplet). The plot includes only statistics with absolute Z-scores below 15 on all sites. A linear trend fitted to the data and lines representing ± 2 SE are shown in red. The intercept of the linear trend was set at 0.
(PDF)

**S15 Fig.  Scatterplots illustrating the effects of 10 ascertainment schemes on Z-scores (a) or $f_4$-statistics (b) of two classes:** 854,358 distinct statistics including up to three African groups and/or up to four East Asian groups, pooled with 749,700 distinct statistics including American/Siberian and/or European and/or Papuan groups. Instead of individual points, heatmaps illustrating point density are shown. Z-scores on all sites (from ca. 5.3 to 15 million sites, as indicated on the x-axes) are compared to Z-scores on ascertained datasets on the y-axes. Ascertainment schemes and site counts are shown in plot titles. The 2nd and 4th rows of plots in each panel represent close-up views on the origin of the plots: they are based on statistics with absolute Z-scores below 15 on all sites. Linear trends fitted to the data and lines representing ± 2 SE are shown in red. Squared Pearson correlation coefficients ($R^2$) for these data are shown on the y-axes. First, linear trends were fitted to the sets of Z-scores or $f_4$-statistics lying near the origin (having absolute Z-score below 15 on all sites), and then the same linear equations were re-fitted to complete sets of Z-scores or $f_4$-statistics. Since we show both $f_4$-statistics and Z-scores side by side here, $R^2$ was used instead of residual SE as a measure of correlation.
(PDF)

**S16 Fig.  The effects of ascertaining SNPs polymorphic in archaic humans on real (a) and simulated data (b, c).** We focused on two $f_4$-statistic classes that are most strongly affected by any type of ascertainment on real data: $f_4$(African$_x$, Neanderthal or Denisovan; African$_y$, non-African) and $f_4$(non-African$_x$, Neanderthal or Denisovan; African, non-African$_y$). On real

data, statistics from these classes were sampled randomly and were calculated on AT/GC sites and on archaic-ascertained sites (transitions and transversions), using all sites without missing data at the level of each quadruplet (i.e., using the "*allsnps = TRUE*" or "*useallsnps: YES*" setting). Papuans and Australians were excluded from the pool of AMH groups due to their Denisovan ancestry component, which was not simulated; and Africans with substantial non-African ancestry (S1 Table) were also removed to make the distinction between various classes of statistics clearer. Archaic ascertainment was performed either on a group composed of the Altai Neanderthal and Denisovan (panel **a**, left), or Vindija Neanderthal and Denisovan (panel **a**, right). A slightly different protocol was used for archaic ascertainment in other parts of this paper since it was performed on a group composed of both Neanderthal individuals and the Denisovan. Graphs illustrating five classes of simulated demographic histories are shown in panel **b** and scatterplots illustrating the effects of ascertaining SNPs polymorphic in the group composed of one "Neanderthal" and one "Denisovan" individual on genetic data simulated according to those histories are shown in panel **c**. The same "Neanderthal" and "Denisovan" individuals were used for ascertainment and for calculating $f_4$-statistics, which is a non-optimal (Fig 4A) but inevitable approach in practice. On the graphs (**b**), the following abbreviations are used: Afr., Africans; nAfr., non-Africans; Den., Denisovan; Neand., Neanderthal. Alternative positions of the out-of-Africa bottleneck simulated at 65, 70, or 75 kya (generation time = 25 years) are marked with red dots. Gene flows from ghost unsampled lineages are shown in green, and those from sampled lineages are shown in blue. Divergence or split times are shown in kya, and effective population sizes are not shown for clarity (see S13 Table). In the first sub-panel (**b**, Model 1) parameters that are the same across all five simulations are shown, and they are omitted from the other sub-panels (**b**, Models 2-5). Results for 68 simulated histories are presented in panel **c**, and by "history" we assume a combination of simulated admixture graph topology, admixture proportions, effective population sizes, population divergence and bottleneck times. The following key simulation parameters are shown in plot titles for Models 1 through 4: model name, proportion of "super-archaic" or Neanderthal-related admixture in ancestral AMH ("Parc"), proportion of Neanderthal or ghost AMH admixture in non-Africans ("PnAfr"), out-of-Africa bottleneck date in kya ("BN"), diploid effective population size during the out-of-Africa bottleneck ("Ne"). The following additional simulation parameters are shown in plot titles for Model 5: proportion of AMH admixture in the Neanderthal lineage ("Pneand"), proportion of Neanderthal admixture in non-Africans ("P1nAfr"), proportion of "ghost AMH" admixture in non-Africans ("P2nAfr"). (PDF)

**S17 Fig. The effects of ascertaining SNPs polymorphic in archaic humans on real (a) and simulated data (b, c).** We focused on 15 $f_4$-statistic classes that are most strongly affected by archaic ascertainment on simulated data (see a list of classes in the legend for panel **a**). On real data, statistics from these classes were sampled exhaustively and were calculated on AT/GC sites and on archaic-ascertained sites (transitions and transversions), using all sites without missing data at the level of each quadruplet (i.e., using the "*allsnps = TRUE*" or "*useallsnps: YES*" setting). Papuans and Australians were excluded from the pool of AMH groups due to their Denisovan ancestry component, which was not simulated; and Africans with substantial non-African ancestry (S1 Table) were also removed to make the distinction between various classes of statistics clearer. Archaic ascertainment was performed either on a group composed of the Altai Neanderthal and Denisovan (panel **a**, left), or Vindija Neanderthal and Denisovan (panel **a**, right). A slightly different protocol was used for archaic ascertainment in other parts of this paper since it was performed on a group composed of both Neanderthals and the Denisovan. Graphs illustrating five classes of simulated demographic histories are shown in panel **b**

and scatterplots illustrating the effects of ascertaining SNPs polymorphic in the group composed of one "Neanderthal" and one "Denisovan" individual on genetic data simulated according to those histories are shown in panel **c**. The same "Neanderthal" and "Denisovan" individuals were used for ascertainment and for calculating $f_4$-statistics, which is a non-optimal (Fig 4A) but inevitable approach in practice. On the graphs (**b**), the following abbreviations are used: Afr., Africans; nAfr., non-Africans; Den., Denisovan; Neand., Neanderthal. Alternative positions of the out-of-Africa bottleneck simulated at 65, 70, or 75 kya (generation time = 25 years) are marked with red dots. Gene flows from ghost unsampled lineages are shown in green, and those from sampled lineages are shown in blue. Divergence or split times are shown in kya, and effective population sizes are not shown for clarity (see S13 Table). In the first sub-panel (**b**, Model 1) parameters that are the same across all five simulations are shown, and they are omitted from the other sub-panels (**b**, Models 2-5). We focused on 15 $f_4$-statistic classes that are most affected by archaic ascertainment on simulated data (see a list of classes in the legends for each panel). Results for 68 simulated histories are presented in panel **c**, and by "history" we assume a combination of simulated admixture graph topology, admixture proportions, effective population sizes, population divergence and bottleneck times. The following key simulation parameters are shown in plot titles for Models 1 through 4: model name, proportion of "super-archaic" or Neanderthal-related admixture in ancestral AMH ("Parc"), proportion of Neanderthal or ghost AMH admixture in non-Africans ("PnAfr"), out-of-Africa bottleneck date in kya ("BN"), diploid effective population size during the out-of-Africa bottleneck ("Ne"). The following additional simulation parameters are shown in plot titles for Model 5: proportion of AMH admixture in the Neanderthal lineage ("Pneand"), proportion of Neanderthal admixture in non-Africans ("P1nAfr"), proportion of "ghost AMH" admixture in non-Africans ("P2nAfr").
(PDF)

**S1 Table. A list of present-day and ancient human genomes analyzed in this study.**
(XLSX)

**S2 Table. A list of non-simulated SNP sets (ascertainment schemes) analyzed in this study and their sizes.**
(XLSX)

**S3 Table. Performance of ascertainment schemes explored across 12 population quintuplets and assessed as the fraction of all admixture graph topologies that are accepted under ascertainment (fit well with WR <3 SE) but rejected on all sites (fit poorly with WR >3 SE).** We also applied the binary classifier to determine if an ascertainment scheme produces unbiased or biased results (the latter cases are highlighted in bold and underlined text). The numbers of population quintuplets or ascertainment schemes affected by bias (according to this classifier) are shown in the rightmost column and in the bottom row, respectively. The composition of the population sets is shown above the table in an abbreviated way: *arch*, archaic humans, followed by the number of archaic groups; *afr*, Africans, followed by the number of African groups; *nafr*, non-Africans or Africans with substantial non-African admixture [67], followed by the number of such groups. The SNP counts correspond to sites polymorphic in larger collections of groups from which the analyzed population quintuplets were taken, see S2 Table. SNP counts vary across the population sets, and minimal and maximal values are shown in separate columns.
(XLSX)

**S4 Table. Performance of ascertainment schemes explored across 12 population quintuplets and assessed as squared Pearson correlation coefficient ($R^2$) for worst $f_4$-statistic**

residuals (WR) of admixture graphs fitted to unascertained vs. ascertained data, based on 5,000 graphs that are best-fitting according to log-likelihood (LL) scores on all sites, or based on all graphs. We also applied the binary classifier to determine if an ascertainment produces unbiased or biased results (the latter cases are highlighted in bold and underlined text). Median $R^2$ values across all population quintuplets or ascertainment schemes are shown in the second rightmost column and in the bottom row, respectively, and the numbers of population quintuplets affected by bias (according to this classifier) are shown in the rightmost column. The composition of the population sets is shown above the table in an abbreviated way: *arch*, archaic humans, followed by the number of archaic groups; *afr*, Africans, followed by the number of African groups; *nafr*, non-Africans or Africans with substantial non-African admixture [67], followed by the number of such groups. The SNP counts correspond to sites polymorphic in larger collections of groups from which the analyzed population quintuplets were taken, see S2 Table. The same results based on LL scores are shown in S5 Table. SNP counts vary across the population sets, and minimal and maximal values are shown in separate columns.
(XLSX)

**S5 Table. Performance of ascertainment schemes explored across 12 population quintuplets assessed as squared Pearson correlation coefficient ($R^2$) for log-likelihood scores (LL) of admixture graphs fitted to unascertained vs. ascertained data, based on 5,000 graphs that are best-fitting according to log-likelihood (LL) scores on all sites, or based on all graphs.** We also applied the binary classifier to determine if an ascertainment produces unbiased or biased results (the latter cases are highlighted in bold and underlined text). Median $R^2$ values across all population quintuplets or ascertainment schemes are shown in the second rightmost column and in the bottom row, respectively, and the numbers of population quintuplets affected by bias (according to this classifier) are shown in the rightmost column. The composition of the population sets is shown above the table in an abbreviated way: *arch*, archaic humans, followed by the number of archaic groups; *afr*, Africans, followed by the number of African groups; *nafr*, non-Africans or Africans with substantial non-African admixture [67], followed by the number of such groups. The SNP counts correspond to sites polymorphic in larger collections of groups from which the analyzed population quintuplets were taken, see S2 Table. The same results based on WR of admixture graphs are shown S4 Table. SNP counts vary across the population sets, and minimal and maximal values are shown in separate columns.
(XLSX)

**S6 Table. Performance of ascertainment schemes on simulated data (those shown in Fig 3) explored across three population quintuplets** (including either "Denisovan", or "Neanderthal 1", or "Neanderthal 2" "archaic" individuals, in addition to the "African 1", "African 2", "non-African 1", and "non-African 2" groups) and assessed as the fraction of all topologies that are rejected under ascertainment (fit poorly with WR >3 SE) but accepted on all sites (fit well with WR <3 SE), or as the fraction of all topologies that are accepted under ascertainment (WR <3 SE) but rejected on all sites (WR >3 SE). We also applied the binary classifier (based on a 10th percentile threshold and 10 random site subsamples matching the average size of the HO one-panel set, 500K sites) to determine if the ascertainment produces unbiased or biased results (the latter cases are highlighted in bold and underlined text). Ten independent simulations with the same parameters were performed, and the following ascertainment schemes were explored on each of them: 1) archaic ascertainment (1.05M sites on average across simulation iterations); 2) HO one-panel ascertainment, based on the "African 2" group (500K sites on average); 3) HO four-panel ascertainment (based on the "African 1",

"African 2", "non-African 1", and "non-African 2" groups, 1.34M sites on average); 4) AFR MAF ascertainment, that is restricting to sites with MAF >5% in the union of the "African 1" and "African 2" groups (1.85M sites on average); 5) global MAF ascertainment on the union of the "African 1", "African 2", "non-African 1", and "non-African 2" groups (1.62M sites on average, abbreviated as "AMH MAF"); 6) non-African MAF ascertainment (1.48M sites on average).
(XLSX)

**S7 Table. Performance of ascertainment schemes explored across 27 exhaustively sampled $f_4$-statistic classes and assessed as residual standard deviation of a linear trend fitted to a scatterplot of Z-scores on unascertained vs. ascertained data (abbreviated as "residual SE" and expressed in the same units as $f_4$-statistic Z-scores).** Only $f_4$-statistics having absolute Z-scores <15 SE (on all sites) were considered. We also applied the binary classifier to determine if an ascertainment produces unbiased or biased results (the latter cases are highlighted in bold and underlined text). The numbers of $f_4$-statistic classes or ascertainment schemes affected by bias are shown in the rightmost column and in the bottom row, respectively. For each $f_4$-statistic class, total number of distinct statistics and the number of statistics having absolute Z-scores <15 SE (on all sites) are shown. The composition of the $f_4$-statistic classes is shown above the table in an abbreviated way: AFR, African populations; AMRSIB, Native American and Siberian populations; ARCH, archaic human individuals (Neanderthals and Denisovans); chimp, chimpanzee; EAS, East Asian populations; EUR, European populations; ME, Mediterranean and Middle Eastern populations; PAP, Papuan and Australian populations. The SNP counts correspond to sites polymorphic in larger collections of groups from which the analyzed population quintuplets were taken, see S2 Table. SNP counts vary across the population sets, and minimal and maximal values are shown in separate columns.
(XLSX)

**S8 Table. Performance of ascertainment schemes explored across 27 exhaustively sampled $f_4$-statistic classes and assessed as squared Pearson correlation coefficient ($R^2$) for $f_4$-statistics calculated on unascertained vs. ascertained data.** Only $f_4$-statistics having absolute Z-scores <15 SE (on all sites) were considered. We also applied the binary classifier to determine if an ascertainment produces unbiased or biased results (the latter cases are highlighted in bold and underlined text). The numbers of $f_4$-statistic classes or ascertainment schemes affected by bias are shown in the rightmost column and in the bottom row, respectively. For each $f_4$-statistic class, total number of distinct statistics and the number of statistics having absolute Z-scores <15 SE (on all sites) are shown. The composition of the $f_4$-statistic classes is shown above the table in an abbreviated way: AFR, African populations; AMRSIB, Native American and Siberian populations; ARCH, archaic human individuals (Neanderthals and Denisovans); chimp, chimpanzee; EAS, East Asian populations; EUR, European populations; ME, Mediterranean and Middle Eastern populations; PAP, Papuan and Australian populations. The SNP counts correspond to sites polymorphic in larger collections of groups from which the analyzed population quintuplets were taken, see S2 Table. SNP counts vary across the population sets, and minimal and maximal values are shown in separate columns.
(XLSX)

**S9 Table. Values of three selected $f_4$-statistics and corresponding Z-scores across eight ascertainment schemes and on unascertained data (all sites).** Five ascertainment schemes featured in S10–S12 Tables are highlighted in bold.
(XLSX)

**S10 Table. Dissecting the statistic $f_4$(Altai Neanderthal, Biaka; Mbuti, Saharawi) belonging to the (archaic, African$_x$; African$_y$, non-African) class.** Sites were stratified by DAF in Africans or DAF in Europeans into three bins: nearly fixed ancestral (DAF < = 5%), non-fixed (DAF 5–95%), and nearly fixed derived (DAF > = 95%). This was done for unascertained data (all sites), for random ascertainment (AT/GC sites), and for four non-random ascertainment schemes as indicated in the leftmost column. The number and proportion of $f_4$-informative sites falling into each DAF bin are shown. Mean DAF in four populations, mean differences in DAF between populations 1 and 2, populations 3 and 4, mean products of the DAF differences (i.e., $f_4$-statistics) and their Z-scores are shown for these frequency bins.
(XLSX)

**S11 Table. Dissecting the statistic $f_4$(Fulani, Juǀ'hoan North; Igbo, Ogiek) composed of four African groups.** Sites were stratified by DAF in Africans or DAF in Europeans into three bins: nearly fixed ancestral (DAF < = 5%), non-fixed (DAF 5–95%), and nearly fixed derived (DAF > = 95%). This was done for unascertained data (all sites), for random ascertainment (AT/GC sites), and for four non-random ascertainment schemes as indicated in the leftmost column The number and proportion of $f_4$-informative sites falling into each DAF bin are shown. Mean DAF in four populations, mean differences in DAF between populations 1 and 2, populations 3 and 4, mean products of the DAF differences (i.e., $f_4$-statistics) and their Z-scores are shown for these frequency bins.
(XLSX)

**S12 Table. Dissecting the statistic $f_4$(Burmese, Dinka; Juǀ'hoan North, Sengwer) composed of three African groups and one East Asian group.** Sites were stratified by DAF in Africans or DAF in Europeans into three bins: nearly fixed ancestral (DAF < = 5%), non-fixed (DAF 5–95%), and nearly fixed derived (DAF > = 95%). This was done for unascertained data (all sites), for random ascertainment (AT/GC sites), and for four non-random ascertainment schemes as indicated in the leftmost column. The number and proportion of $f_4$-informative sites falling into each DAF bin are shown. Mean DAF in four populations, mean differences in DAF between populations 1 and 2, populations 3 and 4, mean products of the DAF differences (i.e., $f_4$-statistics) and their Z-scores are shown for these frequency bins.
(XLSX)

**S13 Table. Parameters of the simulated demographic histories that are not shown on the respective graphs in Figs 3A, S16B and S17B: effective population sizes and sampling or divergence dates.**
(PDF)

**S1 Text. Effects of ascertainment on real data: fits of complex admixture graphs.**
(PDF)

**S2 Text. An overview of $f_4$-statistic biases caused by ascertainment.**
(PDF)

**S3 Text. Mechanisms of bias in selected $f_4$-statistics.**
(PDF)

**S4 Text. Archaic ascertainment explored on simulated genetic data.**
(PDF)

## Acknowledgments

The authors are grateful to Mark Lipson and Nick Patterson for discussions.

## Author Contributions

**Conceptualization:** Pavel Flegontov, Robert Maier, David Reich.

**Data curation:** Pavel Flegontov.

**Formal analysis:** Pavel Flegontov, Ulaş Işıldak, Eren Yüncü, Piya Changmai.

**Funding acquisition:** Pavel Flegontov, Piya Changmai, David Reich.

**Investigation:** Pavel Flegontov, Ulaş Işıldak.

**Methodology:** Pavel Flegontov, Robert Maier.

**Resources:** David Reich.

**Software:** Ulaş Işıldak, Robert Maier.

**Supervision:** Pavel Flegontov, David Reich.

**Visualization:** Pavel Flegontov, Ulaş Işıldak, Eren Yüncü.

**Writing – original draft:** Pavel Flegontov, David Reich.

**Writing – review & editing:** Pavel Flegontov, Ulaş Işıldak, Robert Maier, Eren Yüncü, Piya Changmai, David Reich.

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
