## [Decision Letter · Decision Letter 0]

25 Apr 2023

Dear Dr Flegontov,

Thank you very much for submitting your Research Article entitled 'Modeling of African population history using f-statistics can be highly biased and is not addressed by previously suggested SNP ascertainment schemes' to PLOS Genetics.

The manuscript was fully evaluated at the editorial level and by independent peer reviewers. The reviewers appreciated the attention to an important topic but identified some concerns that we ask you address in a revised manuscript.

We therefore ask you to modify the manuscript according to the review recommendations. Your revisions should address the specific points made by each reviewer.

Yours sincerely,

Charleston Wen-Kai Chiang, Ph.D.

Guest Editor

PLOS Genetics

Xiaofeng Zhu

Section Editor

PLOS Genetics

As all three reviewers pointed out, the authors conducted an exhaustive analysis to investigate and characterize the ascertainment bias and its impact on inferring evolutionary history of African populations. All reviewers also have suggested that the manuscript as currently written is difficult to follow, both at the level of the text and the figure. They have made a number of suggestions to improve the readability of the manuscript. A revision with improved readability for a readership with general knowledge of genetics will be considered.

Reviewer's Responses to Questions

**Comments to the Authors:**

Reviewer #1: Flegontov et al present a very detailed study of the impact of ascertainment on f4 statistics. This is an interesting and important study because f4 statistics are very commonly used in population genetic demography reconstruction in humans and many other taxa. They're particularly commonly used with ancient DNA, which is often sequenced based on a SNP-capture approach, due to the low quality data available. The authors are particularly concerned about ancient DNA from arid and hot regions, such as Africa, where shotgun ancient DNA is particularly unlikely to succeed and capture approaches are especially necessary.

Unfortunately, one of the authors primary findings is that the commonly used 1240K capture array is particularly bad at analyzing African populations. The authors do an absolutely heroic amount of work to explore a number of different ascertainment schemes both using empirical data and simulations. This leads them to having a wide range of suggestions depending on the data at hand, but also pointing out that ascertained data may ultimately make it very difficult to analyze some kinds of African population history---it's very difficult to alleviate the bias with any kind of ascertainment scheme.

One question I have regarding the authors suggestions for future research is whether they are suggesting creating a new SNP array, or subsetting an existing one based on the schemes described here, or some mixture of the two. Not a major point per se, but I'd just like some clarity from the authors.

I don't find the analyses presented to be objectionable, and they largely seem to make sense to me. My most major concern with the manuscript is that I suspect it will be very hard to read for someone who isn't deeply versed in the lore of f-statistics. I think that for general readability and to reach a broader audience, it would be useful if every analyses had a brief introduction to explain exactly what the f-statistic represents and how it helps answer the question at hand. I think that will also help readers gain a little bit more familiarity with what is going on and provide a little more intuition as to what might be going on.

For myself, I don't understand the analysis described on line 449. What's the predictor and the response variable in the linear trend for f4 statistic z-scores? I suspect this is a thing a reader that is intimately familiar with f4 statistics will understand immediately but will be lost on a more general audience. Similarly, it might be worth noting that the f4 statistic for treeness (line 425) is basically the ABBA-BABA test that detected Neandertal admixture in humans, to help orient readers.

Overall, I do really like the paper and it's an amazing amount of work. I'm always happy to see more thorough explorations of the properties of f-statistics, as I feel they were somewhat under explored in early work. However, the readability of the manuscript leaves a lot to be desired, and I suggest the authors make an effort to make the manuscript a bit more accessible to people aren't experts in f-statistics.

I have a couple of other minor comments.

1) The authors use a criterion on the size of the "worst residuals" (WR) from a graph to determine whether to reject the graph or not. Are the properties of this approach known? Does the test have the same size for all topologies with the same number of leaves, or does the internal structure of a topology impact the size of the test? I suppose my concern is that there may be different power and size for different topologies, making it somewhat difficult to interpret the exhaustive enumeration over topologies that the authors performed.

2) Fig 1 is very complicated. I do think it actually conveys a lot of useful information, but it's pretty hard to parse. I think one easy fix is to try to move the 1240K WR example panel out of the middle. It took me a long time to realize that that panel wasn't "special" and was just like the other panels outside the middle.

3) I wonder if Fig 2e can be simplified by presenting it with a log scale on the y axis? That way there may not be a need to have 3 different panels with different y axes, which makes it very weird to look at.

4) Figure 3 seems to just be missing.

I prefer to sign my reviews. My name is Joshua Schraiber

Reviewer #2: The authors present a major study comparing how different ascertainment schemes affect the results given by f4 statistics and reconstructed admixture graphs when including genomic data of individuals from Africa. The manuscript shows the impact of using different ascertainment schemes on whole-genome sequencing data from SGDP, ancient humans and archaic hominins. These analysis show that some ascertainment schemes, such as using polymorphic variants based on data from three archaic hominins, give poorer results compared to other ascertainment schemes that include using particular panels or using variants that are common on many populations in Africa. Then, the authors perform simulations under a realistic demographic scenario to show that there are few biases in the inferred admixture graph when using a scheme similar to the Human Origins SNP panel or using a scheme that includes common variants from Africa.

This is a manuscript that describes an important problem to investigate the past history of Africa. The paper describes very well thought experiments and the authors have made a great effort to understand the impact of ascertainment schemes on the inference of admixture graphs. There are a couple of things that I think would be helpful for readers of this paper: 1) Describe with a little bit more detail the main metric used to analyze how the ascertainment schemes bias demographic reconstructions. Explaining with detail what are “the worst f4-statistic residuals (WR) for graphs” would be useful for readers to assess why particular ascertainment biases could be problematic for studies of past population history. 2) I think it would be very helpful to use consistent abbreviations and notations in the Figures and manuscript that identify all the ascertainment schemes used in the analysis performed in the real data. This would make it easier to follow the results presented by the authors. I would also suggest using a notation and abbreviation for the simulation section that is consistent with the section on the analysis of real data.

Additionally I have a few comments on the manuscript.

Line 115.- “However, evidence is accumulating that supports archaic admixture in Africans (Chen et al. 2020, Hubisz et al. 2020)” This paper contests that claim ( https://www.biorxiv.org/content/10.1101/2022.03.23.485528v3.abstract ). I think it would be good to mention it since it provides good evidence contesting that sentence.

Lines 164-167.- Can the authors briefly explain what the motivation is behind using this ascertainment scheme?

Legend on Supp Fig. 1.- “Suppl. Fig. 1. Scatterplots illustrating the effects of the 1240K ascertainment on LL and WR for exhaustive collections of simple admixture graphs.” It would be useful for the reader to define LL and WR in the text before pointing to this figure. It would be good to give a clear explanation of LL on the main text.

Line 190-193.- “the worst f4-statistic residuals (WR) for graphs including one archaic human, three African groups, and one African group with ca. 60% of non-African ancestry (Fan et al. 2019) are poorly correlated on all sites and 1240K sites (R = 0.31-0.35).” Can you devote more space to explain the metric WR? This metric is key on the first set of analysis and the manuscript would be easier to follow if it is defined explicitly.

“Suppl. Fig. 2. Two alternative approaches for visualizing the effect of ascertainment bias on admixture graph fits illustrated using one population combination, “Denisovan, Khomani San, Mbuti, Dinka, Mursi”.” What are the two alternative approaches? I think it would be good to mention this very explicitly.

Figure 1.- I find the admixture graphs added on the figure not particularly helpful. I do not think they are helpful to understand the main point of this Figure.

Figure 3 is missing from the manuscript.

Reviewer #3: The paper presents an important and comprehensive study of bias caused by SNP ascertainment for commonly used statistics and inference methods in human (ancient) population genetics, specifically methods to measure and interpret human population structure in light of human history.

I think the study is extremely thorough, and the content certainly important enough for the readership of this journal, in my view.

The text is well written and (mostly) guides the reader sufficiently through the complex experiments and results. I have not much to criticise about the actual experiments and content of the study.

My main criticism concern the Figures, some of which are too overloaded and not sufficiently clear. What follows are suggestions on how to improve them.

Figure 1:

I think the chaotic layout does not work and obfuscates the main figure. My suggestion would be to make the main figure (R-squared of WR against various ascertainment panels) panel a), and then place selected WR scatter plots (perhaps three representative ones ?) orderly below as panels b-d. Panels e-g could then be the three trees that are shown. I am not sure whether they are needed though. In any case, _all_ sub-panels must respect a minimum font size of 5 for any text in them, when rescaled to a full width figure. To link the WR-scatter plots with the points in panel a, the authors used purple lines. I suggest to remove them for clarity, and instead mark the selected points with little numbers or symbols which could then be used in the scatter plots and trees to link them with panel a.

Caption: The terms WR and LL need to be introduced right in the caption. I think it is I not sufficient to just introduce these key terms in the main text. It is already challenging enough to go through all the technical details, and it should not be made harder by having to look up these terms first when looking at the figure for the first time. I think introducing them once in the Caption of Figure 1 may be sufficient for the subsequent figures as well.

As an additional comment on this subject: I like Supplementary Figure 1, which tells a clear story and is quick to understand. Perhaps one should consider moving it into the main text.

Figure 2:

My general comment on Figure 2 is to separate it into multiple figures. It's just too much, and they describe separate enough things to justify splitting them apart. My suggestion is to put 2a+b into one figure, and c-f into another one.

Specific comments on subpanels of Figure 2:

a) The dozen or so dotted lines are a distraction, and I would suggest considering simply a linear, or partially linear, or logarithmic time scale on the y axis. The actual year numbers (which are given in unnecessary 4-5 significant digits) can be better included into Supplementary Table 13, which already lists the population sizes.

b) Too small. Again, check the font sizes, they should not be smaller than 5-6pt when scaled on full-width. Perhaps move the legend above the plots, next to panel a) or so, to have more width for the 8 charts. One could also consider ditching too of the trees or so, for example the ones without Neanderthal admixture, which could then simply be described as a special case of the trees with admixture.

c) I could not follow panel c at all. Neither in the caption nor in the main text it is sufficiently explained what the labels on the left mean. For example, "HO-1 non-OG groups (all asc. Inds)" is cryptic and not explained. When the figure is first introduced in the text, it says in lines 366-368 "As illustrated by distributions of true admixture graph WRs in Fig. 2c, ‘blindly’ ascertaining on individuals or sets of groups randomly sampled across the graph almost guarantees rejecting the true historical model by a wide margin". If that is referring to the two red box plots labeled "HO-1 OG (one ind.)" and "HO-1 non-OG groups (one ind.)", that is pure guess-work on my end, because it is not explained. The caption lists various ascertainment schemes for panel c) but with no obvious link to the abbreviations used in the labels. Please expand this, perhaps it gets easier when splitting up that figure as I suggested.

d) is fine.

e) I found a bit unclear why it is needed and what it tells me. What I see is that the derived allele frequency spectra for the "true root" are i) substantially different from any spectrum in the leaves, and ii) that the leave spectra are a lot noiser than the root spectrum. But there is very little difference between the differently grouped leave spectra, and in my view the text doesn't really help to guide the reader through this figure. I think if the authors would like to keep this panel as a main figure, the text needs to describe it properly and tell the reader what to look at and what how to interpret it.

f) is fine.

Figure 3:

Was completely missing from the version of the article that was made available to me. I could not review it.

Figure 4:

The titles of the three panels use the naming scheme introduced in the text, for example AFR2, ARCH 1, ME1. In the caption, this is translated to more concrete F4-statistic-classes. I would suggest to use the F4-form from the caption also as title in the plots to make things easier for the reader.

Figure 5:

Same point as with Figure 4: I suggest to avoid things like F4(AFR3,ARCH1) and instead write F4(AfrX, AfrY; AfrZ, Archaic) or so in the title.

More minor comments:

Supp Fig 1: Again, please explain what WR and LL is. Those abbreviations are explained only in the main text, and in fact after the first mention of Supp Fig 1. Also, the legend of Supp Fig 1 mentions "LL" while I think that is not actually shown, is it?

Table 1: Since this is a main table: Why showing so many different population quintuplets? Aren't three or four representative quintuplets sufficient? And do we need three digits in percent numbers?

Fig 2a: In the caption, it would help if we could know what 'd', 'n', 'a' and so on stand for.

L 665 "one individual sampled at the end of the simulation" What does "end of the simulation" mean here?

Suppl Fig 17b. Too cluttered in my view. Consider moving at least the population sizes into a table?

**Have all data underlying the figures and results presented in the manuscript been provided?**

Reviewer #1: Yes

Reviewer #2: Yes

Reviewer #3: Yes

PLOS authors have the option to publish the peer review history of their article (what does this mean?). If published, this will include your full peer review and any attached files.

Reviewer #1: **Yes: **Joshua G. Schraiber

Reviewer #2: No

Reviewer #3: No

---

## [Decision Letter · Decision Letter 1]

21 Aug 2023

Dear Dr Flegontov,

We are pleased to inform you that your manuscript entitled "Modeling of African population history using f-statistics is biased when applying all previously proposed SNP ascertainment schemes" has been editorially accepted for publication in PLOS Genetics. Congratulations!

Yours sincerely,

Charleston Wen-Kai Chiang, Ph.D.

Guest Editor

PLOS Genetics

Xiaofeng Zhu

Section Editor

PLOS Genetics

Comments from the reviewers (if applicable):

The reviewers generally agreed that this is an important topic for the field of aDNA research; I agree as well. As one of the reviewers pointed out, the text is still dense and as the authors have other similar types of investigations on aDNA toolkits / analysis methods in the work, I would encourage the authors to generally better contextualize the research in the Introduction and Discussion, while clearly but concisely lay out the logic and expectation of each analysis to improve the general readability of the manuscript. Please work with the copy editing team to ensure that high resolution images are submitted for the final publication.

Reviewer's Responses to Questions

**Comments to the Authors:**

Reviewer #1: I am very happy with the authors updated manuscript. I think they've added a lot more context in the introduction and discussion, as well as significantly improved the readability throughout. I think it's still a dense paper, but I don't know that there's much that can be done about that: it's a dense topic! Nonetheless, it's very clear to me that this manuscript will be of extremely high value to those in the field of human ancient DNA, and I hope that it will get people working in ancient DNA in other taxa to realize the importance of ascertainment schemes and careful analyses.

I prefer to sign my reviews. My name is Joshua Schraiber.

Reviewer #2: The authors have addressed all my comments well.

Reviewer #3: The authors have addressed all the comments diligently, and the paper has much improved. May main comments were on the Figures, and I went through them and the legends, and I think they work much better now.

Note that the resolution of the figures as I saw them as a reviewer was unacceptably low. I don't understand why vector graphics weren't used. The TIFFs that had been uploaded were poor resolution!

I have no further comments.

**Have all data underlying the figures and results presented in the manuscript been provided?**

Reviewer #1: Yes

Reviewer #2: Yes

Reviewer #3: Yes

PLOS authors have the option to publish the peer review history of their article (what does this mean?). If published, this will include your full peer review and any attached files.

Reviewer #1: **Yes: **Joshua Schraiber

Reviewer #2: No

Reviewer #3: No

**Data Deposition**

http://datadryad.org/submit?journalID=pgenetics&manu=PGENETICS-D-23-00086R1

**Press Queries**

---

## [Editor Report · Acceptance letter]

1 Sep 2023

PGENETICS-D-23-00086R1 

Modeling of African population history using f-statistics is biased when applying all previously proposed SNP ascertainment schemes 

Dear Dr Flegontov, 

We are pleased to inform you that your manuscript entitled "Modeling of African population history using f-statistics is biased when applying all previously proposed SNP ascertainment schemes" has been formally accepted for publication in PLOS Genetics! Your manuscript is now with our production department and you will be notified of the publication date in due course.

With kind regards,

Dorothy Lannert

PLOS Genetics

On behalf of:
